# Reproductive Tract Microbial Transitions from Late Gestation to Early Postpartum Using 16S rRNA Metagenetic Profiling in First-Pregnancy Heifers

**DOI:** 10.3390/ijms25179164

**Published:** 2024-08-23

**Authors:** Shaked Druker, Ron Sicsic, Shachar Ravid, Shani Scheinin, Tal Raz

**Affiliations:** 1Koret School of Veterinary Medicine, Robert H. Smith Faculty of Agriculture, Food and Environment, The Hebrew University of Jerusalem, Rehovot 7610010, Israel; shaked.druker@mail.huji.ac.il (S.D.); ron.sicsic@mail.huji.ac.il (R.S.); shachar.ravid@mail.huji.ac.il (S.R.); shani.scheinin@mail.huji.ac.il (S.S.); 2Hachaklait, Mutual Society for Veterinary Services, Caesarea Industrial Park, Caesarea 3079548, Israel; 3Advanced Academic Programs, Krieger School of Arts and Sciences, Johns Hopkins University, Baltimore, MD 21218, USA

**Keywords:** 16S rRNA next-generation sequencing, uterus, vagina, pregnancy, primigravida, cow

## Abstract

Studies in recent years indicate that reproductive tract microbial communities are crucial for shaping mammals’ health and reproductive outcomes. Following parturition, uterine bacterial contamination often occurs due to the open cervix, which may lead to postpartum uterine inflammatory diseases, especially in primiparous individuals. However, investigations into spatio-temporal microbial transitions in the reproductive tract of primigravid females remain limited. Our objective was to describe and compare the microbial community compositions in the vagina at late gestation and in the vagina and uterus at early postpartum in first-pregnancy heifers. Three swab samples were collected from 33 first-pregnancy Holstein Friesian heifers: one vaginal sample at gestation day 258 ± 4, and vaginal and uterine samples at postpartum day 7 ± 2. Each sample underwent 16S rRNA V4 region metagenetic analysis via Illumina MiSeq, with bioinformatics following Mothur MiSeq SOP. The reproductive tract bacterial communities were assigned to 1255 genus-level OTUs across 30 phyla. Dominant phyla, accounting for approximately 90% of the communities, included Proteobacteria, Firmicutes, Actinobacteria, Bacteroidetes, and Fusobacteria. However, the results revealed distinct shifts in microbial composition between the prepartum vagina (Vag-pre), postpartum vagina (Vag-post), and postpartum uterus (Utr-post). The Vag-pre and Utr-post microbial profiles were the most distinct. The Utr-post group had lower relative abundances of Proteobacteria but higher abundances of Bacteroidetes, Fusobacteria, and Tenericutes compared to Vag-pre, while Vag-post displayed intermediate values for these phyla, suggesting a transitional profile. Additionally, the Utr-post group exhibited lower bacterial richness and diversity compared to both Vag-pre and Vag-post. The unsupervised probabilistic Dirichlet Multinomial Mixtures model identified two distinct community types: most Vag-pre samples clustered into one type and Utr-post samples into another, while Vag-post samples were distributed evenly between the two. LEfSe analysis revealed distinct microbial profiles at the genus level. Overall, specific microbial markers were associated with anatomical and temporal transitions, revealing a dynamic microbial landscape during the first pregnancy and parturition. These differences highlight the complexity of these ecosystems and open new avenues for research in reproductive biology and microbial ecology.

## 1. Introduction

The intricate interplay between microbial communities and mammals’ reproductive physiologic and pathologic dynamics has garnered significant attention in recent years [1,2,3]. Notably, the transition from late pregnancy to the early postpartum period, characterized by the opening of the cervix, presents a scenario where substantial uterine bacterial contamination commonly occurs, either from the vaginal microbiota or perhaps from the surrounding environment [4,5]. This phenomenon is particularly pertinent for primigravid females, as this is their first ever parturition. In dairy cows, postpartum uterine contamination has been documented in nearly 100% of animals [6,7,8], and the same was suggested in humans and other mammals [1,9,10,11]. Moreover, in dairy cows and women, postpartum uterine inflammatory diseases are more prevalent among primiparous females than multiparous females [12,13,14]. In dairy cows, traditional culture-dependent microbiological analyses have identified several microbial species in the postpartum uteri, including *Escherichia coli*, *Trueperella pyogenes*, *Fusobacterium necrophorum*, *Prevotella melaninogenica*, *Bacteroides* spp., *Pseudomonas* spp., *Streptococcus* spp., and *Staphylococcus* spp., some of which have been associated with metritis [6,7,15,16,17]. However, recent years have witnessed advancements in culture-independent investigations of bacterial communities within postpartum bovine uteri, mainly next-generation sequencing (NGS) of 16S-rDNA amplicon libraries [2,7,8,18]. These approaches have yielded findings that differ from culture-based studies and have enabled the identification of bacteria that are challenging to cultivate through traditional methods. Furthermore, these studies demonstrated that Koch’s postulates are not fulfilled in postpartum uterine diseases [19].

Current research has primarily examined postpartum uterine microbial contamination in dairy cows, particularly concerning metritis. Nevertheless, there is a significant gap in our understanding regarding the transition of microbial communities from late pregnancy to early postpartum, especially in the context of first-pregnancy heifers. This is particularly noteworthy, not just because of the high susceptibility of primiparous cows to uterine inflammatory diseases, but also because the first pregnancy presents a distinctive scenario, offering an unparalleled opportunity to investigate the intricate dynamics of reproductive tract physiology and pathology, along with the environment, in what is probably the first most dramatic contamination of the female reproductive tract. Accordingly, the objective of this study was to describe and compare the community compositions in the vagina at late gestation and in the vagina and uterus at early postpartum, in first-pregnancy heifers using r16S amplicon NGS data.

## 2. Results

A total of 99 samples were obtained from 33 first-pregnancy heifers. Heifers included in the study were those with singleton pregnancies, normal spontaneous vaginal delivery, and no history of pregnancy-related illness during gestation or antibiotic treatments within six weeks before sampling. Moreover, heifers were excluded if they had retained fetal membranes or signs of postpartum septic metritis, as detailed in Section 4.1. Three swab samples were collected from each animal: one from the vagina (‘Vag-pre’) at a median gestation day of 258 (range 254–262 days), and two postpartum samples, from the vagina (‘Vag-post’) and the uterus (‘Utr-post’), at 8 days (median) after calving (range 5–9 days). The DNA from each sample was used for metagenetic analysis of the V4 hypervariable region of the 16s rRNA gene via the Illumina MiSeq platform [20,21,22]. Bioinformatic analysis was performed predominantly according to Mothur MiSeq SOP [23]. In this analysis, a total of 3,887,476 sequences were initially identified, of which, 3,772,145 sequences (97%) remained after screening based on length criteria. Following deduplication, 2,214,729 unique sequences were identified and aligned to the customized SILVA reference database targeting the V4 region of the 16S rRNA gene. Then, an additional screening step resulted in 1,966,847 unique sequences (total number of sequences = 3,175,973), which were filtered to remove overhangs at both ends and underwent another deduplication step, resulting in 1,966,162 unique sequences. Following denoising by pre-clustering and chimera removal, 1,463,757 unique sequences (total number of sequences = 3,090,757) were selected to be classified taxonomically. After identifying and removing undesirable non-bacterial sequences, 1,404,486 high-quality unique sequences remained (total number of sequences = 2,985,052), with an average read length of 291 bp.

The high-quality unique sequences were assigned to 1255 genus-level OTUs. All OTUs were taxonomically assigned to the bacteria domain and were from 30 phyla, 88 classes, 193 orders, 413 families, and 1255 genera. At the per-sample level, the vast majority of samples had >10,000 sequences assigned to OTUs; however, 19 samples had a low sequence count (≤210). Examination of the rarefaction curves (Figure 1) demonstrated that these samples were unrepresentative, and they were excluded from further analyses. Samples that remained in the downstream analyses had a high number of sequences (mean ± SEM: 37,302 ± 1760, Median: 41,244, Range: 10,240–78,602), and their rarefaction curves reached a plateau, indicating an adequate sequencing depth. The number of sequences was significantly higher among samples in the Vag-pre and Vag-post groups, compared to the Utr-post group (41,080 ± 2540, 41,080 ± 3397, 27,722 ± 2578 sequences, respectively; *p* = 0.0007, Kruskal–Wallis One-Way AOV test). Wherever relevant, downstream analyses were performed with a subsampling of 10,000 sequences from each sample.

The bacterial community compositions by phylum in individual samples are presented in Figure 2. The mean bacterial community compositions by phylum in the Vag-pre, Vag-post, and Utr-post groups are shown in Figure 3 and detailed in the corresponding Figure 4. Regardless of the group, the five most dominant phyla by mean relative abundance were Proteobacteria, Firmicutes, Actinobacteria, Bacteroidetes, and Fusobacteria, accounting for approximately 90% of the taxonomically assigned community components. As listed in Figure 4, a comparison of mean bacterial community composition according to phylum revealed significant differences, mainly between the Vag-pre and Utr-post groups, particularly in the phyla Proteobacteria, which was lower, and Bacteroidetes, Fusobacteria, and Tenericutes, which were higher in the Utr-post group. The Vag-post group had relative abundance values that were commonly in between the values of the Vag-pre and Utr-post groups.

The sample coverage, number of observed OTUs, and inverse Simpson diversity estimate parameters (average, 95% CI width, and std) were evaluated for each sample. The analysis was standardized by subsampling 10,000 sequences from each sample 1000 times, and the results were compared among the Vag-pre, Vag-post, and Utr-post groups. As illustrated in Figure 5, the sample coverages were all above 99%, indicating that all samples are highly representative. The mean sample coverage was slightly but significantly higher in the Utr-post group (99.57 ± 0.03%, 99.56 ± 0.03%, 99.73 ± 0.03%; *p* = 0.0002). The mean number of defined OTUs did not differ between the Vag-pre and Vag-post groups; however, it was significantly lower in the Utr-post group (221.7 ± 8.4, 189.1 ± 11.6, 103.7 ± 15.8 OTUs, *p* < 0.0001). Furthermore, the inverse Simpson diversity estimate parameters (average, 95% CI width, and std), also known as 1/D, did not differ between the Vag-pre and Vag-post groups but were significantly lower in the Utr-post group (1/D average: 11.8 ± 0.8, 12.7 ± 1.1, 6.9 ± 1.3, *p* < 0.0001; 1/D 95% CI width: 0.91 ± 0.07, 0.84 ± 0.08, 0.38 ± 0.08, *p* < 0.0001; 1/D Std: 0.19 ± 0.02, 0.18 ± 0.02, 0.08 ± 0.02, *p* < 0.0001). Overall, these results indicated that the postpartum uterus had lower bacterial richness and diversity compared to the vagina before and after parturition.

The non-metric multidimensional scaling (NMDS) ordination [24] of the bacterial communities from the Vag-pre, Vag-post, and Utr-post samples is shown in Figure 6 (3-dimensions; Stress = 0.197; Coefficient of Determination, R^2^ = 0.77). To explore the observed spatial separation of community structures, the AMOVA test was applied, indicating that the centers of the clouds representing each group significantly differ from each other (*p* < 0.001). The HOMOVA test analysis revealed that the bacterial community variations among samples in the Vag-pre and Vag-post groups did not differ (*p* = 0.099), but it was significantly higher in the Utr-post group (SSwithin/(Ni−1) values = 0.170, 0.208, 0.325, respectively, *p* < 0.001; Bonferroni pairwise error rate = 0.0167).

The comparative analysis among the Vag-pre, Vag-post, and Utr-post groups, utilizing non-parametric LEfSe analysis, uncovered distinct microbial profiles associated with each anatomical niche (Figure 7, Figure 8 and Figure 9, Appendix A). In the comparison between Vag-pre and Vag-post, 65 discriminant features were identified (Figure 7, Appendix A). Notable genera with the highest logMaxMean values (>4) and LDA scores (>3.5) in the Vag-pre group included *Trinickia*, *Micrococcaceae_unclassified*, *Intrasporangiaceae_unclassified*, *Staphylococcus*, and *Betaproteobacteria_unclassified*, among others. On the other hand, Vag-post exhibited higher logMaxMean values for *Fusobacterium*, *Leptotrichiaceae_unclassified*, *Histophilus*, *Mycoplasmopsis*, *Parvimonas*, *Bacteroidales_unclassified*, and *Fusobacteriaceae_unclassified*, among others. When comparing Vag-pre with Utr-post, 113 discriminant features were uncovered (Figure 8, Appendix A). Significant top genera in the Vag-pre group included *Trinickia*, *Micrococcaceae_unclassified*, *Intrasporangiaceae_unclassified*, *Ruminococcaceae_unclassified*, *Glutamicibacter*, *Micrococcales_unclassified*, and *Streptococcus*, among others; while Utr-post showed higher logMaxMean values and LDA scores for *Gemella*, *Leptotrichiaceae_unclassified*, *Trueperella*, and *Streptobacillus*, among others. Lastly, in the comparison between the postpartum vagina and uterus, 71 discriminant features were identified (Figure 9, Appendix A). Genera such as *Trinickia*, *Leptotrichiaceae_unclassified*, *Micrococcaceae_unclassified*, *Ruminococcaceae_unclassified*, *Fusobacteriaceae_unclassified,* and *Clostridiales_unclassified* had the highest logMaxMean values and LDA scores in the Vag-post group, while *Trueperella*, *Streptococcus*, and *Peptoniphilus* were more prominent in the Utr-post group. The distinctive genera identified in each of these comparisons provide insights into the specific microbial dynamics associated with these anatomical niches.

The probabilistic Dirichlet Multinomial Mixtures (DMM) model analysis revealed a minimum Laplace value at K = 2 (Figure 10), indicating that this unsupervised analysis segregates the samples into two distinct community types based on their inherent community structures [25]. Of the 80 samples included in the analysis, 34 were allocated to one community type (community P1) and 46 to a second community (community P2). Interestingly, the vast majority of the samples from the Utr-post group (20/24, 83.3%) were segregated to community type P1, the vast majority of samples from the Vag-pre group (29/32, 90.6%) were segregated to community type P2, while the samples from Vag-post were divided quite evenly between the two (P1: 11/24, 45.8%; P2: 13/24, 54.2%).

The first 36 genera that were most responsible for separating the samples into the two community types in the DMM model analysis are presented in Figure 11. This list of top genera represents the cumulative 50% proportion of the total abundance. Community type P1 predominantly includes the majority of Utr-post samples (83.3%), with notable genera such as *Histophilus* (2.73%), *Helcococcus* (2.77%), *Fusobacterium* (5.34%), *Firmicutes_unclassified* (1.39%), *Micrococcaceae_unclassified* (0.67%), *Fusobacteriaceae_unclassified* (1.98%), and *Bacteroides* (1.55%) showing higher relative abundances. In contrast, community type P2 predominantly includes the majority of Vag-pre samples (90.6%), with significantly higher relative abundances of genera such as *Trinickia* (12.82%), *Micrococcaceae_unclassified* (9.11%), *Firmicutes_unclassified* (1.10%), *Ruminococcaceae_unclassified* (2.07%), *Lachnospiraceae_unclassified* (1.51%), *Bacteroides* (0.79%), *Betaproteobacteria_unclassified* (1.33%), and *Peptoniphilus* (1.02%). Key differences include the stark contrast in *Trinickia*, which has a mean abundance of 12.82% in P2 compared to 1.04% in P1, highlighting its association with the Vag-pre group, while *Histophilus* and *Helcococcus* are significantly more abundant in P1, indicating their association with the Utr-post group. Additionally, *Fusobacterium* shows a higher mean abundance in P1 (5.34%) compared to P2 (4.01%), and genera such as *Ruminococcaceae_unclassified*, *Lachnospiraceae_unclassified*, and *Peptoniphilus* are more abundant in P2, suggesting their association with the Vag-pre group. Overall, these findings suggest a distinct microbial profile associated with the prepartum vaginal environment (associated mainly with community type P2), a unique postpartum uterine microbial signature (associated mainly with community type P1), and two communities that imply a transitional phase or variability in the postpartum vaginal microbiome.

## 3. Discussion

The results of this study highlight the distinct transition of microbial communities in the reproductive tract of first-pregnancy heifers, from late pregnancy to early postpartum, focusing on the vagina and uterus. The study addressed a significant gap in understanding the dynamics of microbial colonization during this critical period, as the uterus is substantially contaminated. This is particularly interesting in primiparous cows, as the first parturition is a once-in-a-lifetime event, most likely characterized by the first and most massive microbial invasion into the relatively naïve uterus. Furthermore, our study may provide insights into the increased susceptibility of primiparous females to uterine inflammatory diseases. By employing next-generation sequencing of 16S-rDNA amplicon libraries, we aimed to comprehensively characterize the bacterial communities and shed light on their changes across anatomical sites and time points. Samples were collected from the vagina at gestation day 258 ± 4 and from the vagina and uterus at postpartum day 7 ± 2 to align with our study objectives and the gestation period of Holstein Friesian cows. With an average gestation length of 280 days (typical range of 270–290 days), sampling at day 258 ± 4 was expected to capture the vaginal microbiota before major delivery-associated changes, establishing a baseline for late gestation before the cervix opens [26,27,28]. The postpartum sampling at day 7 ± 2 allowed observation of early microbial colonization and shifts following parturition, a period marked by uterine involution and new microbial establishment. This timing is consistent with several previous studies focused on the postpartum uterine microbiome in dairy cows [8,18,29,30]. Our results revealed notable shifts in microbial composition between the prepartum vagina (Vag-pre), postpartum vagina (Vag-post), and postpartum uterus (Utr-post), indicated by several analytic approaches. Notably, the postpartum uterus exhibited lower richness and diversity compared to the prepartum and postpartum vagina, indicating a dynamic microbial landscape associated with parturition. The lower diversity and richness in the Utr-post group suggest a selective environment in the postpartum uterus, possibly due to the luminal secretions, immune response, or other physiological changes occurring during and after parturition. These findings align with previous findings in dairy cows and women [1,6,8,31,32], suggesting an impact of delivery on uterine microbial colonization.

The distinct endocrine environment during late gestation and early postpartum may have significantly influenced the microbial community composition in the reproductive tract in our study [33]. During late pregnancy, elevated levels of progesterone and estrogen modulate the vaginal and uterine environments, impacting microbial colonization and community structure [33,34,35]. Progesterone, in particular, may promote the growth of certain bacteria while suppressing others, leading to a distinctive microbial profile [36,37,38]. After parturition, there is a dramatic shift in hormonal levels, with a decrease in progesterone and a possible increase in estrogen, which may contribute to changes in microbial communities [6,33,39,40]. The drop in progesterone allows for the initiation of uterine involution, a process that may influence microbial dynamics in the postpartum uterus [41,42,43]. The physiological changes associated with involution, such as increased blood flow and changes in uterine secretions, can alter the microbial habitat, leading to shifts in community composition and diversity [34,42]. Additionally, the postpartum period is marked by a more open cervix and increased exposure to external environmental contaminants, which can introduce new microbial taxa into the reproductive tract [6,43]. Therefore, endocrine regulation should be considered when interpreting our observed microbial diversity and composition shifts between prepartum and postpartum states, warranting further studies.

The meticulous data preparation and sequence quality control steps performed via Mothur were crucial for ensuring the reliability of our results [23,44]. By applying rigorous screening and denoising procedures, we reduced potential biases introduced by sequencing and PCR errors [23,45]. Also, it enabled us to recognize and exclude samples whose data were likely to be unrepresentative, more commonly among postpartum samples. The presence of low-quality samples in both postpartum vagina and uterus, while scarce in the prepartum vagina, may be due to several factors. The increased difficulty in obtaining high-quality samples from the uterus, attributed to its internal location and potential for contamination during collection [46], may contribute to the higher incidence of low-quality uterine samples. Postpartum vaginal and uterine samples might also face challenges during collection, possibly due to residual fluids or tissues affecting DNA extraction [47]. The relatively low occurrence of low-quality prepartum vaginal samples could be associated with less invasive collection methods and a cleaner environment compared to the postpartum period. Additionally, hormonal and physiological changes during and after parturition might influence the microbial composition and DNA quality [48,49,50], further impacting the sample quality in these reproductive tract compartments. Overall, quality control measures enhanced the robustness of our findings and allowed for more accurate interpretations of the observed microbial dynamics.

The analysis of phylum-level relative abundance provided insights into the major players in the microbial communities. Proteobacteria, Firmicutes, Actinobacteria, Bacteroidetes, and Fusobacteria emerged as the dominant phyla, consistent with previous studies in the bovine reproductive tract [2,8,30]. Importantly, significant differences between the Vag-pre and Utr-post groups, particularly in Proteobacteria, Bacteroidetes, and Fusobacteria, suggest a distinct uterine microbial signature postpartum, as also observed in the genus-level analyses.

Alpha diversity metrics, including sample coverage, observed OTUs, and inverse Simpson diversity estimates, provided valuable insights into community richness and diversity [20,51]. The lower diversity in the Utr-post group indicated a shift in the uterine microbial landscape, potentially influenced by postpartum physiological changes and the initiation of uterine involution. These findings underscore the importance of considering both anatomical location and postpartum status when studying microbial diversity. The NMDS ordination plot illustrated spatial separation among Vag-pre, Vag-post, and Utr-post samples, emphasizing the distinct community structures associated with each anatomical niche [52]. AMOVA results further supported significant differences between the groups [53], highlighting the impact of anatomical location on bacterial community variation. The LEfSe analysis identified discriminative genera, providing a finer resolution of microbial differences between groups [54]. Key genera, such as *Trinickia*, *Micrococcaceae_unclassified*, and *Streptococcus*, were more abundant in Vag-pre samples, while *Gemella* and *Trueperella* were enriched in Utr-post samples.

The DMM model analysis revealed two distinct community types, P1 and P2, segregating the samples. These distinct microbial community types identified by the DMM model analysis underscore the complex and dynamic nature of microbial transitions in the reproductive tract during late gestation and early postpartum. Community type P1, predominantly associated with the Utr-post samples, was characterized by higher relative abundances of genera such as *Histophilus*, *Helcococcus,* and *Fusobacterium*. In contrast, Community type P2, which includes most Vag-pre samples, shows significantly higher relative abundances of genera like *Trinickia*, *Micrococcaceae_unclassified*, and *Ruminococcaceae_unclassified*. These distinct microbial profiles suggest that the prepartum vaginal environment (Vag-pre) and the postpartum uterine environment (Utr-post) harbor unique microbial communities, while the postpartum vaginal samples (Vag-post) exhibit transitional characteristics. Identifying specific genera associated with each community type provides valuable insights into the microbial dynamics accompanying anatomical and temporal changes in the reproductive tract of first-pregnancy heifers.

As detailed above, our findings identify specific microbial markers associated with anatomical and temporal transitions. Interestingly, *Trinickia*, a genus of gram-negative, aerobic, and rod-shaped bacteria in the phylum Proteobacteria (recently renamed as Pseudomonadota) [55], was found in high abundance in the prepartum vagina of first-pregnancy heifers. To the best of our knowledge, this genus has not previously been reported in the bovine female reproductive tract; it is a relatively newly recognized genus with limited information available [56,57]. *Histophilus* has been reported in the cow reproductive tract and was associated with high abundances in cows diagnosed with purulent vaginal discharge and metritis [30,58]. The genera *Fusobacterium* and *Trueperella* have been reported extensively in the bovine uterus and have been associated with postpartum uterine diseases [8,58]. A high prevalence of the genus *Gemella* has been reported in the vagina of pregnant women and was associated with alteration of the immune response, preterm delivery, and postpartum uterine complications [59,60]. To the best of our knowledge, *Gemella* has not been reported in the cow reproductive tract but has recently been recognized in high abundance in the seminal microbiota of bull semen [61].

Based on the observed differences in bacterial composition among the Vag-pre, Vag-post, and Utr-post samples, it is plausible that the postpartum uterine bacterial composition results from both vaginal bacterial invasion and potential environmental sources. Significant shifts in microbial profiles from the late-pregnancy vaginal state to the postpartum uterine state suggest that vaginal microbiota may seed the uterine microbiome. The close anatomical proximity between the vagina and uterus supports bacterial migration from the lower to upper reproductive tract. However, some unique microbial signatures in postpartum uterine samples, differing from the vaginal microbiota, indicate that environmental exposures during and after delivery may also shape the uterine microbiota. Additionally, uterine contamination, bacterial community establishment, and involution processes influence uterine secretions, likely affecting postpartum vaginal bacterial dynamics. These factors may explain changes in postpartum vaginal microbiota compared to late pregnancy, while some pre-calving vaginal characteristics remain, albeit modified. Thus, reproductive bacterial compositions are temporally and spatially affected by the unique characteristics of each niche, with different niches (vagina, uterus, and environment) likely influencing each other. The relative prominence of each scenario is challenging to determine definitively without detailed metagenomic analyses (such as Shotgun Metagenomic Sequencing, or Long Read Sequencing), additional functional data, and well-planned environmental sampling [62,63]. Future studies utilizing advanced sequencing techniques and comprehensive environmental monitoring could further elucidate the dynamics of postpartum uterine bacterial colonization and clarify the relative contributions of vaginal and environmental sources.

The implementation of Next-Generation Sequencing technology and advanced analytical methods has significantly enhanced our understanding of microbial communities within the reproductive tract of first-pregnancy heifers. The high-resolution nature of our approach allowed for the identification of subtle changes in community composition, offering a detailed portrayal of the dynamics within these intricate ecosystems [20]. The assignment of OTUs to phylotypes, as conducted in our study, provides a taxonomic framework that aids in discerning the diversity and relationships among microbial populations [45,64]. This method allows for a more nuanced analysis compared to calculating pairwise distances between aligned DNA sequences and clustering OTUs accordingly, as it may account for phylogenetic relatedness, potentially providing a more biologically meaningful representation of microbial diversity [65]. However, this approach has some disadvantages [45]. Assigning OTUs by phylotypes typically relies on sequence similarity thresholds for clustering, which may lead to overestimation or underestimation of species diversity based on the chosen threshold [66,67]. This can introduce biases and affect the accuracy of taxonomic assignments. In contrast, calculating uncorrected pairwise distances provides a more continuous and quantitative measure of sequence similarity, allowing for a more focused assessment of genetic relationships between organisms [66,68]. The use of phylotype-based OTU assignment may result in splitting single taxa into multiple OTUs, depending on the chosen threshold, which might have occurred in our study to some degree. This lack of resolution can limit the precision of microbial community analysis, especially in environments with high species diversity or closely related species [66,67,68]. On the other hand, clustering OTUs based on sequence distances allows for a more flexible and data-driven approach, capturing a finer-scale representation of microbial diversity. However, it is essential to consider the computational demands and potential biases associated with different distance metrics when employing this approach [44,64]. The choice between phylotype-based OTU assignment and sequence distance-based clustering also depends on the research objectives, the nature of the microbial community under investigation, and the available computational resources [66]; all were part of our consideration when the data was analyzed.

There are other inherent limitations associated with our study approach. Potential biases introduced during sample collection and DNA extraction processes may impact the accuracy of our results. Variability in sampling techniques or the DNA extraction process could introduce biases in representing certain taxa, potentially influencing the observed microbial composition. Moreover, while our study contributes valuable insights into the taxonomic aspects of microbial dynamics, further investigations employing functional analyses would be instrumental in elucidating the ecological roles of these microbial communities [62,63]. Future studies should also compare the reproductive tract microbiota before and after calving in both first-pregnancy heifers and multiparous cows and explore how their dynamics may be associated with postpartum uterine diseases. Integrating metagenomic or metatranscriptomic approaches could unveil functional attributes, shedding light on the potential contributions of these microbes to reproductive health and overall host physiology. The key conclusions drawn from the current study are summarized in Section 5.

## 4. Materials and Methods

### 4.1. Animals and Sampling Procedures

All animal handling and sampling procedures were performed by a certified veterinarian in accordance with the ethical approval obtained from the Hebrew University Institute Animal Care and Use Committee (IACUC MD-13-13807-2). Three swab samples were collected from first-pregnancy Holstein Friesian heifers, as detailed below; one from the vagina at gestation day 258 ± 4 (‘Vag-pre’), and two additional samples from the vagina (‘Vag-post’) and uterus (‘Utr-post’) at postpartum day 8 ± 3. Heifers included in the study were from a commercial dairy farm in Israel. They were housed in loose housing systems in large, completely covered open sheds, and fed a total mixed ration, according to the Nutrient Requirement of Dairy Cattle recommendations. All pregnancies were achieved following artificial insemination with thawed frozen semen (fAI) while the mature heifer was on spontaneous estrus, and pregnancy diagnosis was performed by trans-rectal palpation of the uterus 40–50 days after the fAI date (i.e., indicated as the first gestation day) and reconfirmed before the first sampling at late gestation. Individual cow health and reproduction data were recorded using the herd management computer program NOA (Israeli Cattle Breeders Association, Caesarea, Israel). Heifers were excluded from the study if they had a history of pregnancy-related illness during gestation or if they were treated with antibiotics during the six weeks before the first sampling until after three swab samples were collected. Furthermore, heifers were excluded from the study if they had difficult assisted calving, twin pregnancy, postpartum retained fetal membranes, or systemic clinical signs indicating postpartum septic metritis. All heifers included in the study were calved by spontaneous vaginal delivery. Accordingly, the study included data from 99 swab samples collected from 33 first-pregnancy Holstein Friesian heifers.

Swab samples were collected using acceptable methods [8,69]. Before sampling, the vulva was cleaned with a brush and soap water, followed by wipes containing chlorhexidine gluconate and isopropyl alcohol. Sampling was performed using a sterile, disposable, double-guarded endometrial swab (76 cm long; Equivet, Denmark Cat # 290955) inserted into the vagina and managed by one hand while it was directed to the sampling location transrectally by the other hand. For vaginal sampling, the swab was inserted into the cranial vagina, where the inner protective sheath and the swab gradually extruded, and the protruded swab was used to sample the most cranial portion of the vagina and the external os of the cervix for approximately 10 s. For uterine sampling, the protected swab was guided transcervically into the uterine body, where the inner sheath and the swab gradually extruded, and the swab was pressed and rotated against the endometrium for approximately 10 s. Before retraction, the inner swab was inserted back into its protective sheaths to avoid contamination while withdrawing it from the reproductive tract. Outside the reproductive tract, the inner swab tip was kept in its designated sterile sheath and transported at 4 °C to the lab. In the lab, the tip of the swab was placed into a sterile vial containing 1 mL sterile PBS; the vial was vortexed (2 min, maximum speed, twice, 10 min apart) to release the biological material, the tip was discarded, and the fluid was kept frozen at −80 °C until DNA extraction.

### 4.2. DNA Extraction

DNA extraction was carried out using the DNeasy^®^ Blood & Tissue kit (catalog number 69504, Qiagen, Hilden, Germany) according to the manufacturer’s instructions. Extracted DNA was quantified in a Qubit 4 fluorometer using the dsDNA HS and the dsDNA BR assay kits (Thermo Scientific, Waltham, MA, USA). Additional quality evaluation included NanoDrop 2000 spectrophotometer (Thermo Scientific, Waltham, MA, USA) absorbance curve inspection and 260/280 and 260/230 ratio. DNA samples were kept at −20 °C until use.

### 4.3. 16S rRNA Gene Amplicon Libraries Preparation and NGS Sequencing

Genomic DNA was subjected to polymerase chain reaction (PCR) amplification using the 515F and 806R primers, targeting the V4 hypervariable region of the bacterial and archaeal 16S rRNA gene [20,21]. The primers, CS1_515F (ACACTGACGACATGGTTCTACAGTGCCAGCMGCCGCGGTAA) and CS2_806R (TACGGTAGCAGAGACTTGGTCTGGACTACHVGGGTWTCTAAT), were designed with common sequences 1 (CS1) and 2 (CS2) at their 5′ ends, as previously described [22]. Primers were purchased from Integrated DNA Technologies as standard primers and diluted to 10 µM concentration. DNA samples were adjusted to a 10–20 ng/µL concentration before the amplification process. Amplification reactions were conducted in 96-well plates with a 20 µL reaction volume. The master mix for the entire plate included 10 µL of OneTaq Quick-Load 2X Master Mix with standard buffer (New England Biolabs, Ipswich, MA, USA), 0.5 µL of each primer (10 µM), and 8 µL of nuclease-free water. One microliter of template DNA was added to each well. The cycling conditions consisted of an initial denaturation at 95 °C for 5 min, followed by 35 cycles of denaturation at 95 °C for 30 s, annealing at 56 °C for 45 s, and extension at 72 °C for 30 s. A final extension step of 3 min at 72 °C was performed. Each plate included a positive control containing E. coli DNA and a negative control (double-distilled water, DDW). Visible amplification was confirmed through 1.7% agarose gel electrophoresis. Subsequently, PCR products were purified and normalized using the SequalPrep normalization plate kit (Thermo Scientific, Waltham, MA, USA) following the manufacturer’s instructions. Size selection was achieved using Pippin Prep (Sage Science, Beverly, MA, USA).

Following the normalization and size selection steps, a second round of PCR amplification was conducted in 10 µL reactions within 96-well plates. The master mix for the entire plate was prepared, with the composition of each reaction as follows: 5 µL of 2X AccuPrime SuperMix II (Life Technologies, Carlsbad, CA, USA), 2 µL of double-distilled water (DDW), 1 µL of each primer (4 µM), and 1 µL of the PCR product obtained from the previous stage. Each reaction employed a distinct primer pair sourced from the Access Array Barcode Library for Illumina Sequences. Each primer pair featured a unique 10-base barcode (Fluidigm, South San Francisco, CA, USA; Cat # 100-4876). Reactions with unique barcodes were designated for the positive control, no-template control (reaction 1), and a second no-template control reaction containing only Access Array Barcode library primers.

The cycling conditions were programmed as follows: an initial denaturation at 95 °C for 5 min, followed by 8 cycles of denaturation at 95 °C for 30 s, annealing at 60 °C for 30 s, and extension at 68 °C for 30 s. A final elongation step of 7 min at 68 °C concluded the PCR. Validation of the PCR yield for positive and negative controls, along with select samples, was performed using Qubit analysis and size quantification utilizing an Agilent TapeStation 2200 device with D1000 ScreenTape (Agilent Technologies, Santa Clara, CA, USA). Subsequently, samples were pooled in equal volumes and subjected to purification using solid-phase reversible immobilization (SPRI) cleanup with AMPure XP beads at a ratio of 0.6X (*v*/*v*) SPRI solution to sample. Quality control was performed using TapeStation 2200 and Qubit analysis, and the pool was diluted to 6.5 pM for emulsion PCR. Pooled libraries were then sequenced on the Illumina MiSeq platform using V2 reagent kit (150 bp paired-end; UIC Research Resources Center, University of Illinois) [20].

### 4.4. Bioinformatic Analysis

The bioinformatic analysis was performed by using the Mothur program (v.1.48.0) in an interactive mode via Linux terminal, based on the Mothur MiSeq SOP (https://mothur.org/wiki/miseq_sop/, accessed on 13 November 2023) [23]. Paired-end fastq files (R1, R2) from 99 samples obtained from 33 heifers were included in the metagenetic analysis. Data preparation and initial sequence quality control steps to minimize sequencing and PCR errors included read concatenation, contig formation, and length screening. Throughout the process, sequence numbers and their lengths were assessed, and screening parameters were adjusted accordingly to remove sequences based on length criteria. This included the removal of sequences in the upper and lower 2.5th percentiles and adjusting the remaining sequence lengths to ≤293 bp, which is suitable for the V4 region of the 16S rRNA gene. Subsequently, a sequence deduplication step was performed to reduce redundancy (merging duplicates) and streamline downstream analyses. The unique sequences were aligned to the SILVA reference database (Version 132), which was customized to target the V4 region. Coordinates for the V4 region (start = 11895, end = 25318) were set based on a parallel analysis conducted via Mothur by aligning the E. coli rrsB 16S rRNA gene sequence (NCBI Gene ID 948466) against the reference alignment. An additional screening step was applied to remove sequences in the upper and lower 2.5th percentiles to maintain sequences within the desired region. Sequences were then filtered to remove overhangs at both ends, and an additional deduplication step was performed to remove redundancy. Denoising was carried out by pre-clustering using a pseudo-single linkage algorithm developed by Huse et al. (2010), allowing up to two differences between sequences, aiming to remove sequences likely due to pyrosequencing errors [67]. Chimeras were removed by using the VSEARCH algorithm [70]. The sequences were then classified taxonomically using a Bayesian classifier with the mother-formatted RDP training set (Version 18), and non-bacteria reads, such as 16S rRNA from chloroplasts, mitochondria, Archaea, 18S rRNA, and random sequences, were excluded from further analyses. The remaining high-quality sequences were summarized taxonomically and were used for downstream analyses.

The generation of operational taxonomic units (OTUs) was performed from phylotypes, which were classified by genus. Relative abundances were calculated by dividing the read abundance value of an OTU in a specific sample by the total number of reads for that sample. The subsequent steps involved calculating alpha diversity metrics and constructing rarefaction curves, which were used to identify unrepresentative samples that should be excluded from further analyses. The number of sequences, sample coverage, number of observed OTUs, and inverse Simpson diversity estimates were calculated and compared among the groups (Vag-pre, Vag-post, Utr-post).

Beta diversity was assessed using the Bray–Curtis dissimilarity index, and a phylip-formatted distance matrix was used to generate an output file for a non-metric multidimensional scaling (NMDS) plot (carried out in 3 dimensions to achieve a stress value below 0.2) [71]. To explore the spatial separation of community structures observed in the NMDS plot, the molecular variance analysis (AMOVA test) was applied, indicating whether the centers of the clouds representing a group are more separated than the variation among samples of the same group. Furthermore, a homogeneity of molecular variance (HOMOVA test) was conducted to statistically compare variations among the Vag-pre, Vag-post, and Utr-post groups.

The non-parametric LEfSe analysis was conducted to provide additional insights into community composition and identify key microbial taxa distinguishing the different groups [54]. The analysis was performed with a threshold value of 2 on the logarithmic score for the Linear Discriminant Analysis (LDA). Furthermore, the Dirichlet Multinomial Mixtures model was used to calculate Laplace values, which were utilized in an unsupervised evaluation of the partitioning of samples into distinct community types [25]. This analysis was performed to reveal inherent community structures (sample segregation and significant OTUs driving community segregation) without specifying sample groupings.

### 4.5. Additional Statistical Analysis & Plotting

Most statistical analyses were performed using Mothur, as detailed above [23]. However, additional statistical analyses were conducted when applicable using Statistix 10 software (Analytical Software, Tallahassee, FL, USA) and Prism 5.01 (GraphPad Software, San Diego, CA, USA). The choice of statistical tests was guided by the nature of the parameters analyzed to ensure adherence to the assumptions of each test. A non-parametric Kruskal–Wallis one-way analysis of variance (ANOVA) test was employed for non-normally distributed variables, complemented by Dunn’s all-pairwise comparisons test when relevant. Repeated Measures ANOVA or a mixed model was applied to account for dependencies when applicable. Proportional data were compared using Fisher’s exact test. All statistical tests were based on two-tailed hypotheses, with significance established at *p* < 0.05 and adjustments made for multiple comparisons. Tabular presentations were created using Microsoft Excel, while graphical visualizations were generated using Prism 5.01 (GraphPad Software) or Python (Python Software Foundation, Python Language Reference, version 3.10.12). When applicable, data analysis and visualization were performed using in-house custom Python scripts, leveraging the Pandas library for data manipulation, the Matplotlib library for 2D visualizations, and the mplot3d toolkit for generating 3D plots, such as the NMDS plot.

## 5. Conclusions

Our study provides data that support the conclusion that distinct microbial profiles exist within the reproductive tract of first-pregnancy heifers, influenced by anatomical and temporal transitions. Notably, the microbial profiles of the vagina at late gestation and the uterus at early postpartum are the most distinct, with the postpartum vagina serving as an intermediary link between them. At the phylum level, dominant phyla in the reproductive tract include Proteobacteria, Firmicutes, Actinobacteria, Bacteroidetes, and Fusobacteria; however, there is a significant reduction in Proteobacteria and an increase in Bacteroidetes, Fusobacteria, and Tenericutes in the postpartum uterus compared to the late gestation vagina. At the genus level, the late gestation vagina is characterized by higher abundances of *Trinickia* and *Micrococcaceae*, while there is a substantial increase in *Histophilus*, *Helcococcus*, and *Fusobacterium* in the postpartum uterus. Additionally, there is a substantial reduction in bacterial richness and diversity in the postpartum uterus compared to the vagina, indicating a selective niche possibly shaped by endocrine and other physiological changes associated with parturition and early postpartum uterine involution. Our findings support the hypothesis that vaginal microbiota may seed the postpartum uterine microbiome due to their close anatomical proximity. However, unique microbial signatures in the postpartum uterus, differing from vaginal microbiota, imply that environmental exposures during and after delivery may also play a role in shaping the uterine microbiota. Overall, our study highlights the distinct transition of microbial communities in the reproductive tract of first-pregnancy heifers from late pregnancy to early postpartum, emphasizing the need for further research using advanced sequencing techniques to fully understand the functionality of these microbial dynamics and their implications for reproductive physiology and health.

## Figures and Tables

**Figure 1 ijms-25-09164-f001:**
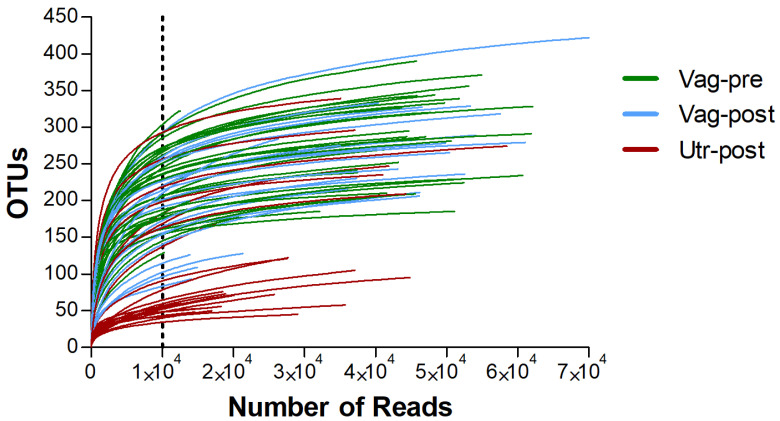
Rarefaction curves of 16S sequencing samples. Curves show the number of identified OTUs (*Y*-axis) as a function of the number of reads (*X*-axis). Samples from the Vag-pre, Vag-post, and Utr-post groups are represented in green, light blue, and red, respectively. Only samples with >10,000 reads (marked by the vertical dotted line) were included in further analyses.

**Figure 2 ijms-25-09164-f002:**
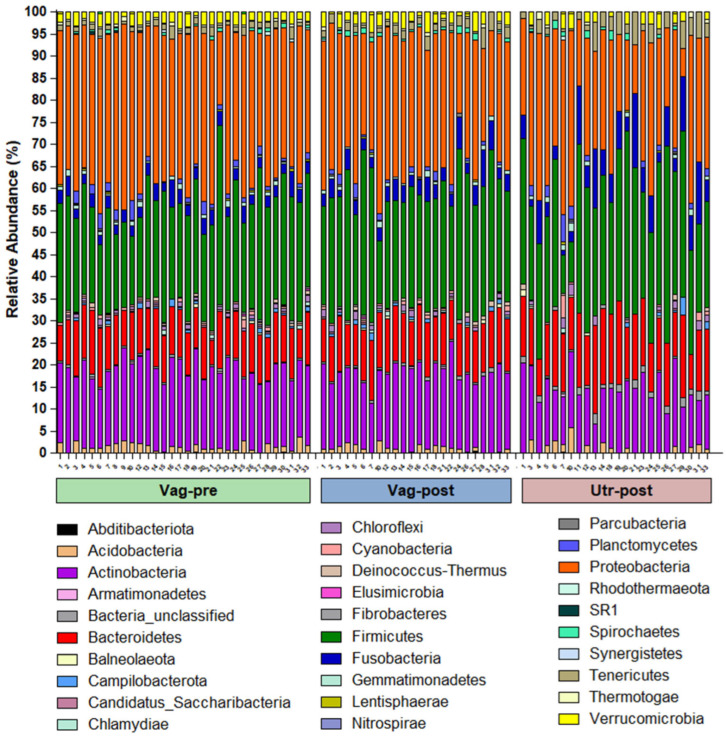
Individual stacked bar plots of bacterial community composition according to phylum. Each bar represents one sample, and the cow identity number is noted below the *X*-axis.

**Figure 3 ijms-25-09164-f003:**
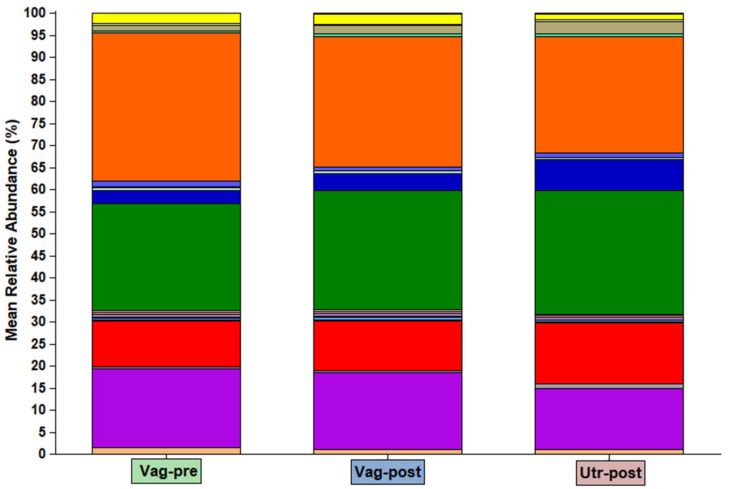
Group stacked bar plots of mean bacterial community composition according to phylum in the Vag-pre, Vag-post, and Utr-post groups. The color of each phylum is presented in Figure 2. The mean relative abundance values and the statistical comparisons among the groups are detailed in Figure 4.

**Figure 4 ijms-25-09164-f004:**
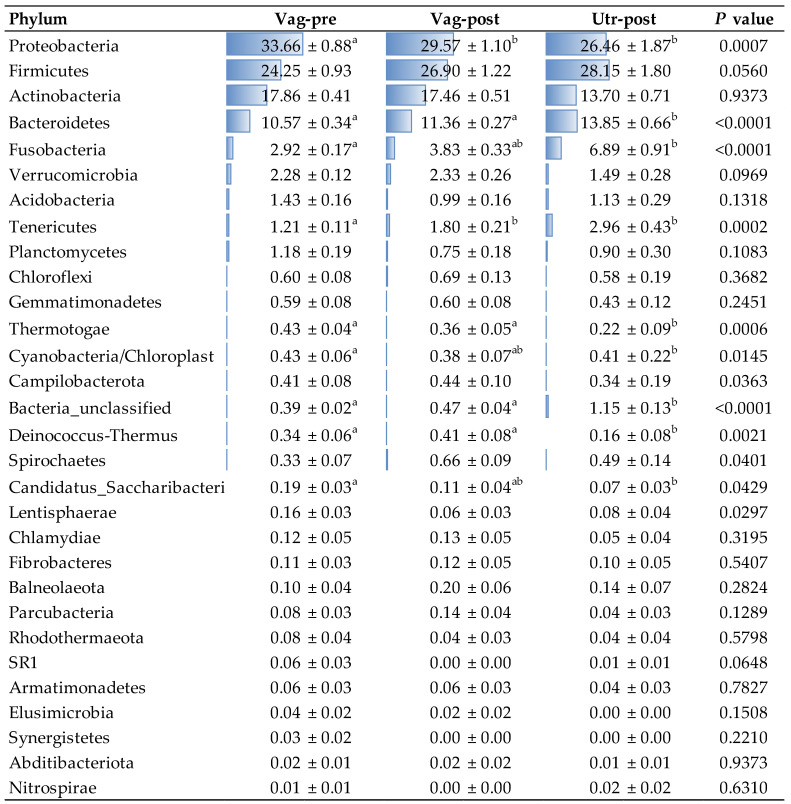
Mean relative abundance of different phyla among the Vag-pre, Vag-post, and Utr-post groups. Data is related to Figure 3. Mean relative abundance (%) is presented as mean ± SEM; Conditional formatting (i.e., blue horizontal column bars) was applied to mean values to visually represent the data and highlight key differences. ^a,b^ Different letters above values within a line represent significant differences between groups.

**Figure 5 ijms-25-09164-f005:**
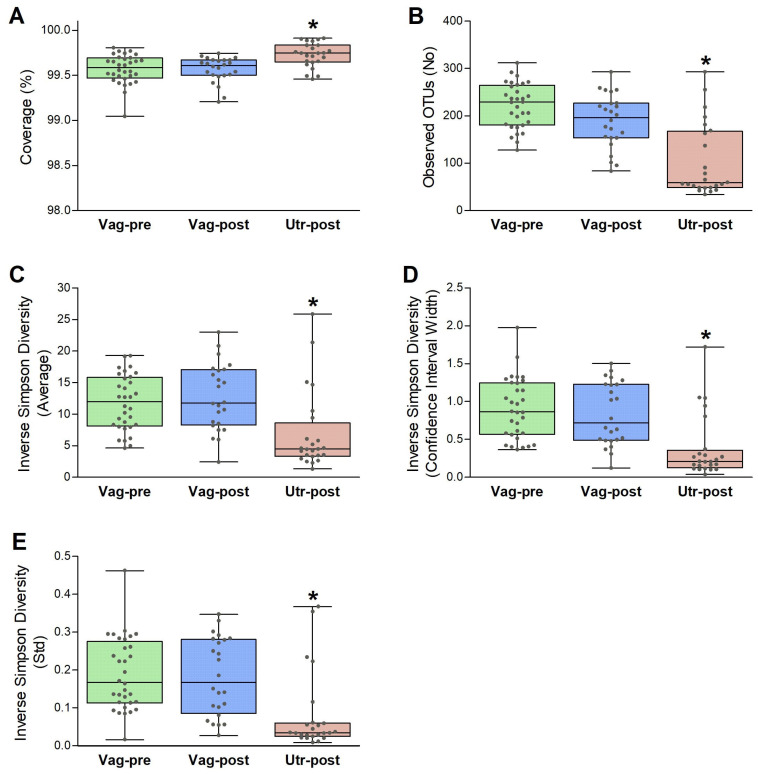
Alpha diversity metrics. (**A**) Sample coverages (%). (**B**) Number of observed OTUs. (**C**–**E**) Inverse Simpson diversity estimate averages (**C**), 95% confidence interval widths (**D**), and standard deviations (**E**). These parameters were evaluated for each sample, with the analysis standardized by subsampling 10,000 sequences from each sample 1000 times [23]. Results were compared among the Vag-pre, Vag-post, and Utr-post groups. Box and whisker plots are presented with the overlap of scatter plots. The median (central black line) is shown in the middle of each box, with the 25% and 75% percentile ranges (box depth) and the maximum and minimum (whiskers); each dot represents a value of a single sample. Kruskal–Wallis one-way AOV, followed by Dunn’s all-pairwise comparisons. * *p* ≤ 0.0002.

**Figure 6 ijms-25-09164-f006:**
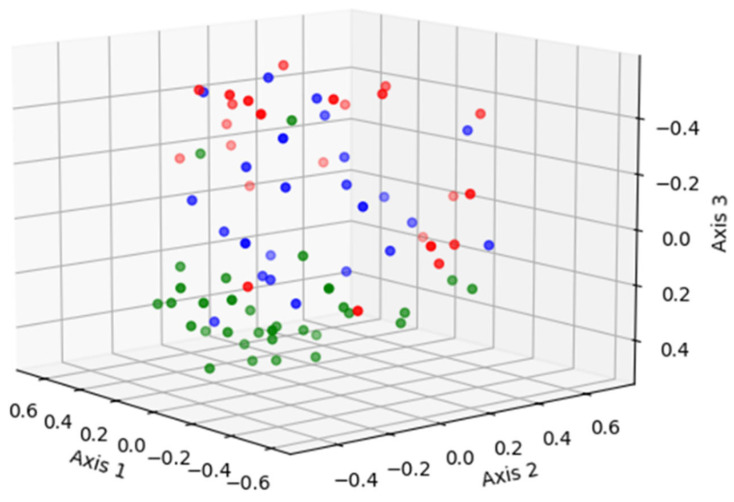
Non-metric Multidimensional Scaling (NMDS) ordination plot of the bacterial community composition from the Vag-pre (green), Vag-post (blue), and Utr-post (red) samples (3-dimensions; Stress = 0.197; R^2^ = 0.77).

**Figure 7 ijms-25-09164-f007:**
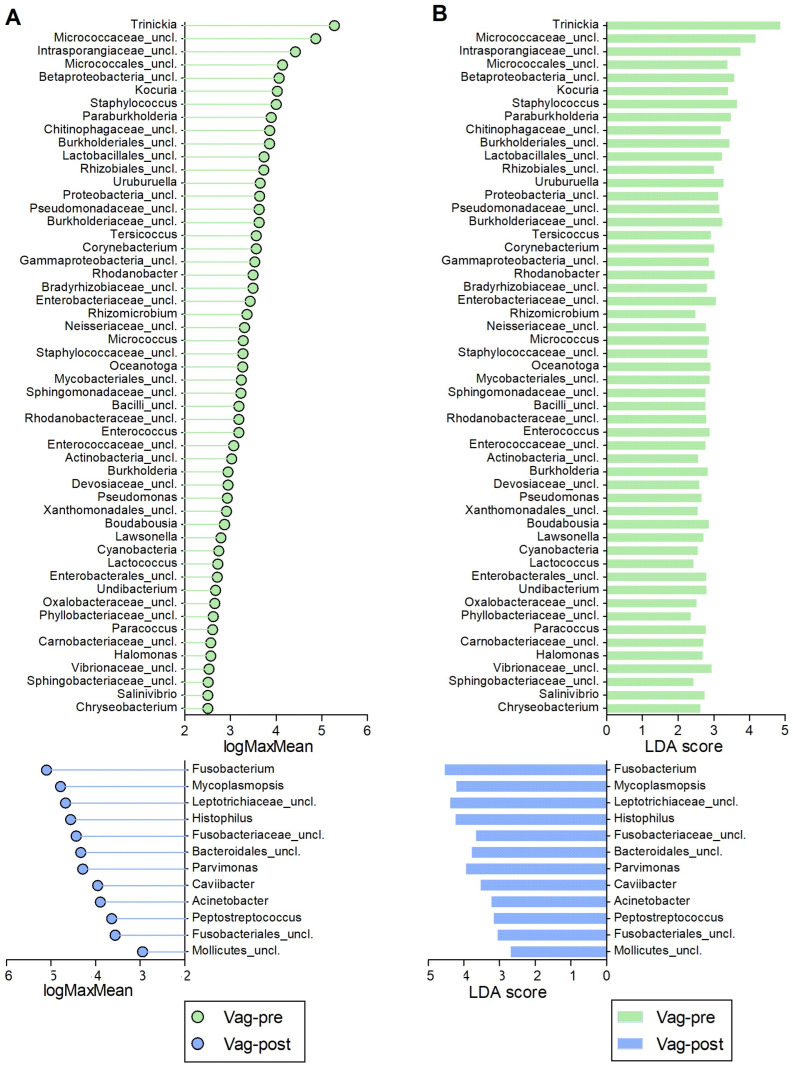
Discriminative genera obtained from LEfSe comparison between Vag-pre and Vag-post samples. (**A**) Lollipop chart of the logMaxMean values, indicating the logarithm value of the maximum mean abundance of the OTU across the specified groups. (**B**) Corresponding bar chart of the Linear Discriminant Analysis (LDA) scores. *p* < 0.05 for all values. Vag-pre is indicated in green, and Vag-post is indicated in blue. Corresponding data are detailed in Appendix A.

**Figure 8 ijms-25-09164-f008:**
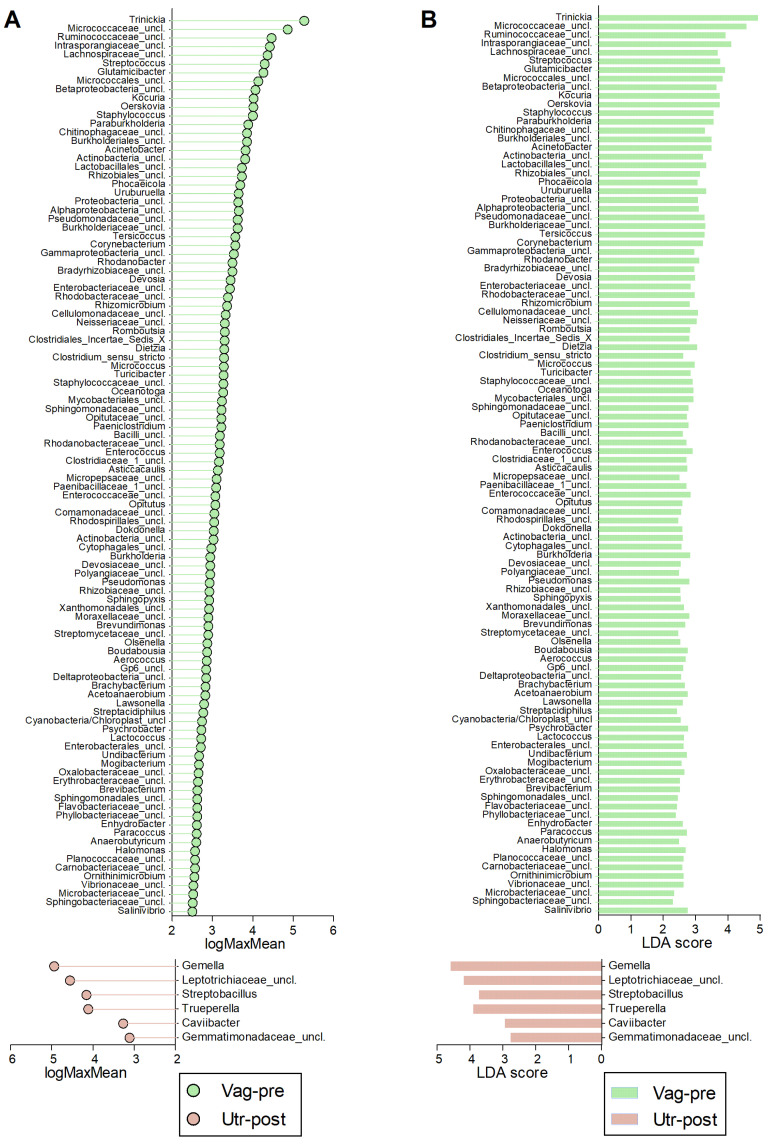
Discriminative genera obtained from LEfSe comparison between Vag-pre and Utr-post samples. (**A**) Lollipop chart of the logMaxMean values, indicating the logarithm value of the maximum mean abundance of the OTU across the specified groups. (**B**) Corresponding bar chart of the Linear Discriminant Analysis (LDA) scores. *p* < 0.05 for all values. Vag-pre is indicated in green, and Utr-post is indicated in red. Corresponding data are detailed in Appendix A.

**Figure 9 ijms-25-09164-f009:**
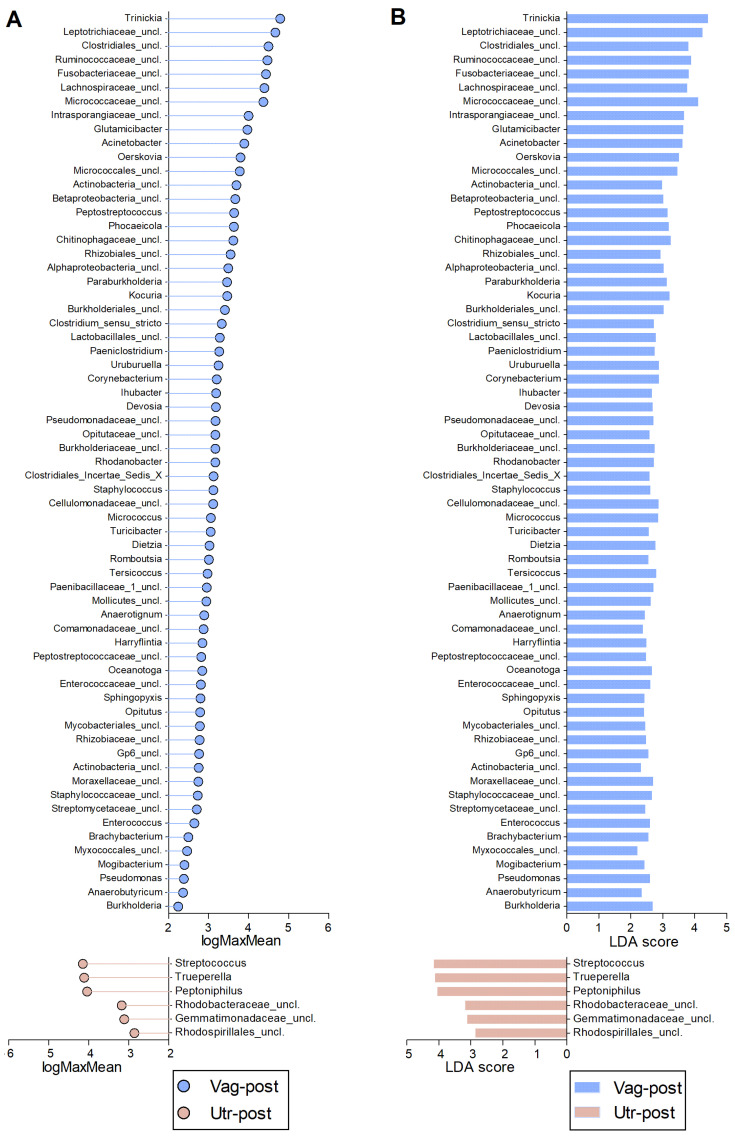
Discriminative genera obtained from LEfSe comparison between Vag-post and Utr-post samples. (**A**) Lollipop chart of the logMaxMean values, indicating the logarithm value of the maximum mean abundance of the OTU across the specified groups. (**B**) Corresponding bar chart of the Linear Discriminant Analysis (LDA) scores. *p* < 0.05 for all values. Vag-post is indicated in blue, and Utr-post is indicated in red. Corresponding data are detailed in Appendix A.

**Figure 10 ijms-25-09164-f010:**
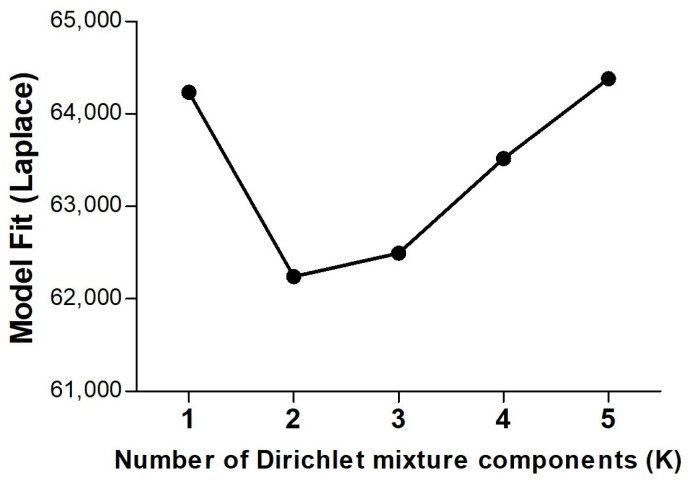
Model fit using the probabilistic Dirichlet Multinomial Mixtures (DMM) model analysis. The *Y*-axis represents the Laplace values as the number of Dirichlet mixture components K increases.

**Figure 11 ijms-25-09164-f011:**
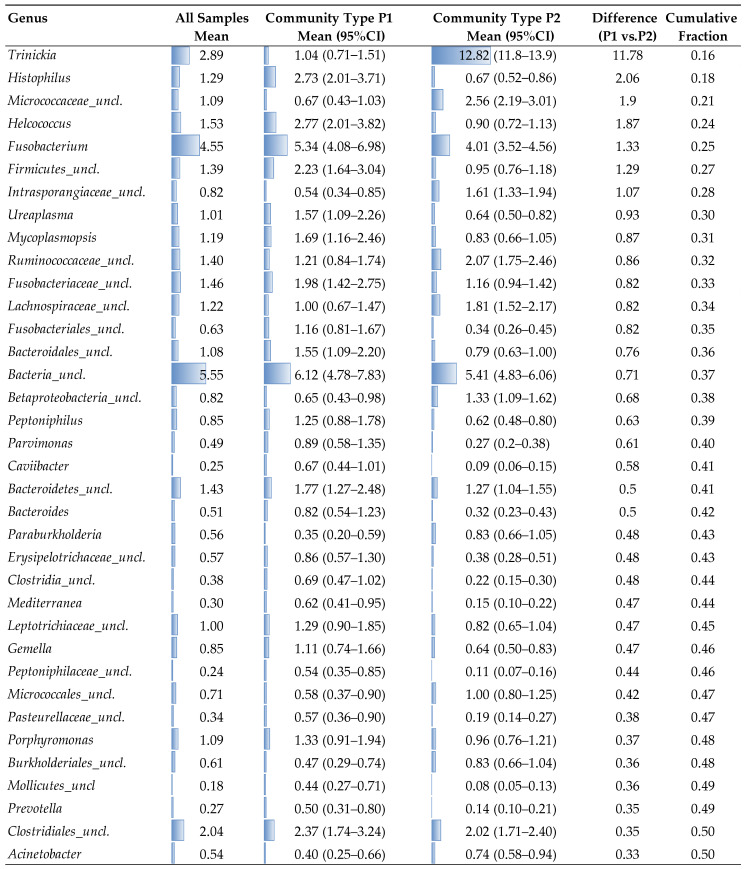
Relative abundance of different genera in two distinct community types (P1 and P2) defined by unsupervised probabilistic Dirichlet Multinomial Mixtures (DMM) model analysis. The DMM analysis segregated most Utr-post samples (83.3%) to community P1, and most Vag-pre samples (90.6%) to community P2, while Vag-post samples were divided between P1 (45.8%) and P2 (54.2%). Relative abundances (%) of the first 36 genera most responsible for separating the samples into these two community types are presented (as mean and 95% CI). Data presented also include the relative abundances of all samples before segregation, the difference between P1’s and P2’s relative abundances, and the cumulative proportion of the total abundance explained by the listed genera. Conditional formatting (i.e., blue horizontal column bars) was applied to mean values to represent the data visually. uncl., unclassified.

## Data Availability

Sequencing data generated during this study have been deposited in the Sequence Read Archive (SRA) of the National Center for Biotechnology Information (NCBI) and are publicly available under BioProject PRJNA1143924.

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
