# Peer review of "Reproductive Tract Microbial Transitions from Late Gestation to Early Postpartum Using 16S rRNA Metagenetic Profiling in First-Pregnancy Heifers"

_ijms, 2024, doi:10.3390/ijms25179164_

Round 1

Reviewer 1 Report

Comments and Suggestions for Authors

This is an interesting study. The authors performed a spatial and temporal characterization of reproductive tract microorganisms in 33 first-pregnant heifers, using 16S rRNA sequencing to describe and compare community composition in the vagina during late gestation and in the vagina and uterus during the early postpartum period in first-pregnant heifers. Justification of the experiment is clear and methodology and statistical procedures were appropriate to fully reach the objective raised. However, some aspects must be completely corrected and clarified before considered it publication.

Abstract:

The presentation of the background in the abstract section is too redundant, and changes to the abstract section are recommended. For example, Line16-19. In addition, the abstract contains few descriptions of the results of this study, especially the variations in the characterization of the microbiota at the phylum and genus level. It is recommended that the authors carefully revise the abstract section.

Keywords:

Suggested deletion “reproductive tract metagenomics”.

Results

The data analyzed in the table can be presented in the text, while for the other parts of the data it is recommended that the table be presented in the form of an annex. If possible, it would be better if the data in the tables were presented in the form of Figures. In addition, it was found that the headings of Table 4 are bolded, while the headings of Table 1, Table 2, Table 3, and Table 5 are not, so it is suggested that the authors should standardize the format of the headings.

Lines 117-128: Phylum level microbes should be non-italicized, please check.

Line 137: The content of the notes on the table in the heading section of the table can be placed after the table in the format of a table note. The same applies to other tables in the text.

Line 205: Table 2 describes genus level microorganisms, so the table should say "Phylum" instead of "genus". The same applies to Table 3 as well as Table 4.

Line 256: Table 5 is only listed in the text and is not described in the text; it is recommended that the author add a description of the relevant section.

Discussion

The discussion section is confusing. Authors need to put a lot of effort into revising this section. This is because it looks more like a review article than a research article. The authors are advised to go through their own experimental data and analyze it through in-depth discussions to reveal certain scientific truths. This study analyzed the characteristics of reproductive tract flora of primiparous heifers at different stages of pregnancy by 16S rRNA sequencing. Therefore, the discussion can be focused on, for example, the changes in the abundance of some specific flora, the impact on the microecological environment of the reproductive tract and further beneficial/harmful effects on heifers. For example, Line 338: the authors describe Trinickia is discussed and the authors should discuss in depth the effects of changes in Trinickia abundance on heifers at different stages of gestation in the context of the results of this study.

Line 347: This paragraph is missing the punctuation mark ".", author is advised to double-check.

Line 277 and Line 416: "16S-rDNA" appears in this sentence, which is inconsistent with the title "16S rRNA", please confirm.

Materials and Methods

Line 594: "P" should be italicized in this paragraph, and "P-Value" should be italicized throughout the article, please check.

Author Response

Dear Reviewer 1,              

Thank you very much for taking the time to review our manuscript and for providing positive feedback and valuable criticism. Based on your and the other reviewers' judgment and comments, the manuscript underwent substantial changes, including the Abstract, Introduction, descriptions of the Methods and Results, as well as relevant matching changes in the Discussion. Please find the detailed responses below and the revisions made. As part of our submission, two main manuscript files were uploaded - a clean copy (uploaded as the main manuscript file), and a copy with track changes (uploaded via Non-published Material).

We hope you will find the revised version suitable for publication in IJMS.

_ _ _

Comment 1: Abstract: The presentation of the background in the abstract section is too redundant, and changes to the abstract section are recommended. For example, Line16-19. In addition, the abstract contains few descriptions of the results of this study, especially the variations in the characterization of the microbiota at the phylum and genus level. It is recommended that the authors carefully revise the abstract section.

Response 1: As recommended, the abstract was substantially revised to eliminate redundancy and increase the descriptions of the results.

_ _ _

Comment 2: Keywords: Suggested deletion "reproductive tract metagenomics".

Response 2: The keyword was deleted as suggested.

_ _ _

Comments regarding Results:

Comment 3: The data analyzed in the table can be presented in the text, while for the other parts of the data it is recommended that the table be presented in the form of an annex. If possible, it would be better if the data in the tables were presented in the form of Figures. In addition, it was found that the headings of Table 4 are bolded, while the headings of Table 1, Table 2, Table 3, and Table 5 are not, so it is suggested that the authors should standardize the format of the headings.

Response 3: Thank you very much for your constructive feedback regarding the presentation of data in the tables. Based on your valuable suggestions, we have standardized the format of the table headings and legends according to the journal guidelines. We have also ensured that the main findings are clearly presented and highlighted in the text, with appropriate references to the corresponding figures or tables.

Recognizing the importance of presenting data clearly and effectively, and as recommended, we have replaced Tables 2, 3, and 4 with corresponding figures (Figures 6, 7, and 8, respectively). To maintain clarity and transparency, and because the original tables contain valuable information that may be of interest to some readers, these three tables have been moved to the supplementary material (Tables S1, S2, and S3, respectively).

Tables 1 and 5 from the original submission have been retained in the revised manuscript (as Table 1 and Table 2). We have improved their appearance, their legends, and the text referring to them. The mean bacterial community compositions according to phylum in the Vag-pre, Vag-post, and Utr-post groups are shown in Figure 3 and detailed in the corresponding Table 1, which provides complementary data.

Regarding Table 5 from the original manuscript (results of the DMM analysis), while we considered converting it into a figure, we found it challenging due to the large volume of data and the detailed nature of the information. We believe that transforming this detailed table into a figure might not effectively convey the complexity and nuances of the results. However, the conditional formatting (in the form of blue horizontal column bars) included in all tables (including Table 2 in the revised version) enhances the visual representation of the data, making key differences more apparent. This method effectively combines elements of both table and figure formats.

We sincerely appreciate your valuable comments regarding the data presentation, and we acknowledge that these improvements have significantly enhanced our manuscript.

_ _ _

Comment 4: Lines 117-128: Phylum level microbes should be non-italicized, please check.

Response 4: Thanks to the reviewer's comment, we ensure that the nomenclature used in our manuscript is consistent with the conventions recommended by the International Committee on Systematics of Prokaryotes (ICSP) for the nomenclature of bacteria (Oren et al., Int. J. Syst. Evol. Microbiol.  2023;73:005585 DOI 10.1099/ijsem.0.005585).

_ _ _

Comment 5: Line 137: The content of the notes on the table in the heading section of the table can be placed after the table in the format of a table note. The same applies to other tables in the text.

Response 5: The heading section in the tables was revised as instructed by the reviewer.

_ _ _

Comment 6: Line 205: Table 2 describes genus level microorganisms, so the table should say "Phylum" instead of "genus". The same applies to Table 3 as well as Table 4.

Response 6: Thanks for spotting this mistake. The tables (which are now Tables S1, S2, and S3) were corrected.  

_ _ _

Comment 7: Line 256: Table 5 is only listed in the text and is not described in the text; it is recommended that the author add a description of the relevant section.

Response 7: As instructed,  the main results presented in Table 5 (which is now Table 2) are described in the Results section and further explained and discussed in the Discussion section.

_ _ _

Comments regarding Discussion:

Comment 8: The discussion section is confusing. Authors need to put a lot of effort into revising this section. This is because it looks more like a review article than a research article. The authors are advised to go through their own experimental data and analyze it through in-depth discussions to reveal certain scientific truths. This study analyzed the characteristics of reproductive tract flora of primiparous heifers at different stages of pregnancy by 16S rRNA sequencing. Therefore, the Discussion can be focused on, for example, the changes in the abundance of some specific flora, the impact on the microecological environment of the reproductive tract and further beneficial/harmful effects on heifers. For example, Line 338: the authors describe Trinickia is discussed and the authors should discuss in depth the effects of changes in Trinickia abundance on heifers at different stages of gestation in the context of the results of this study.

Response 8: The Discussion section was extensively revised based on this comment and relevant comments from other reviewers. The revised discussion now focuses more on our experimental data

_ _ _

Comment 9: Line 347: This paragraph is missing the punctuation mark ".", author is advised to double-check.

Response 9: Corrected as suggested.  

_ _ _

Comment 10: Line 277 and Line 416: "16S-rDNA" appears in this sentence, which is inconsistent with the title "16S rRNA", please confirm.

Response 10: Corrected as suggested (“16S-rDNA" was changed to “16S-rRNA”)

_ _ _

Comments regarding Materials and Methods

Comment 11: Line 594: "P" should be italicized in this paragraph, and "P-Value" should be italicized throughout the article, please check.

Response 11: The format presentation of  P value was corrected throughout the manuscript as instructed by the reviewer.

Thank you!

Reviewer 2 Report

Comments and Suggestions for Authors

This study expands avenues for future research in reproductive biology, microbial ecology, and reproductive tract diseases by comparing differences in microbes and their community composition during three periods of the birth canal in pregnant heifers. This is a well-written manuscript and an interesting study, but the amount of work is too small for publication in this journal.

1.         The title of the manuscript is too long;

2.         The data and pictures presented in the results section are confusing and unorganized;

3.         There is no specific conclusion after the discussion, which is recommended to be added;

4.         The indexes of animal selection are too vague, please supplement the data, and it is suggested to supplement the indexes of serum immunity and inflammatory factor;

5.         The materials and methods section introduces the classification of operational units according to the genus level, and the results section shows the classification at the phylum level, which is recommended to be checked;

6.         According to the preface, the article intends that microorganisms are linked to inflammation or disease, whereas this article only does pre-work microbial analysis and does not clearly express whether or not it screens for key microorganisms associated with inflammation;

7.         The table in the article is not conducive to observing the data and it is recommended to change it to a picture.

Author Response

Dear Reviewer 2,       

Thank you very much for taking the time to review our manuscript and for providing positive feedback and valuable criticism. Based on your and the other reviewers' judgment and comments, the manuscript underwent substantial changes, including the Abstract, Introduction, descriptions of the Methods and Results, as well as relevant matching changes in the Discussion. Please find the detailed responses below and the revisions made. As part of our submission, two main manuscript files were uploaded - a clean copy (uploaded as the main manuscript file), and a copy with track changes (uploaded via Non-published Material).

We hope you will find the revised version suitable for publication in IJMS.

_ _ _

Comment (General): This study expands avenues for future research in reproductive biology, microbial ecology, and reproductive tract diseases by comparing differences in microbes and their community composition during three periods of the birth canal in pregnant heifers. This is a well-written manuscript and an interesting study, but the amount of work is too small for publication in this journal.

Response: Thank you for your feedback on our manuscript. We appreciate your recognition of the significance and potential impact of our study in advancing research in reproductive biology, microbial ecology, and reproductive tract diseases.

Our study is the first to provide a detailed comparative analysis of microbial communities in the reproductive tract of first-pregnancy heifers during two critical periods: late pre-partum (Vag-pre) and early postpartum (Vag-post and Utr-post). By utilizing next-generation sequencing of 16S-rRNA amplicon libraries, we identified significant shifts in microbial composition, revealing distinct microbial profiles and dynamics associated with each period. This novel insight is crucial for understanding the role of microbial communities in reproductive health and disease, particularly in primiparous individuals who are at a higher risk of postpartum uterine inflammatory diseases.

Regarding your comment on the amount of work, it is important to note that our study involved a comprehensive analysis of 99 samples from 33 heifers, providing robust and statistically significant results. The fact that the samples were taken from the same individuals is also a strength of our study. The application of advanced bioinformatics and statistical methods, such as the Dirichlet Multinomial Mixtures (DMM) model and LEfSe analysis, further strengthens the validity of our findings. Therefore, we believe that our study is suitable for publication in the IJMS. It lays a critical foundation for future research and has significant implications for improving reproductive management and health in both veterinary and human medicine. The insights gained from our work can inform the development of targeted interventions to modulate microbial communities and enhance reproductive outcomes.

_ _ _

Comment 1: The title of the manuscript is too long

Response 1: Thank you for your feedback on the title of our manuscript. Based on your suggestion, we have revised the title to be more concise while still accurately reflecting the content and scope of our study. The new title is "Reproductive Tract Microbial Transitions from Late Gestation to Early Postpartum Using 16S rRNA Metagenetic Profiling in First-Pregnancy Heifers." This revised title maintains specificity and relevance, clearly communicating the key aspects of our research. We appreciate your constructive feedback and believe this new title better meets the journal's guidelines for conciseness and relevance.

_ _ _

Comment 2: The data and pictures presented in the results section are confusing and unorganized.

Response 2: Thank you very much for your feedback regarding the organization and presentation of data in the results section. Based on the valuable suggestions provided by reviewers, we have made several improvements to enhance the clarity and organization of the data presented. We have standardized the format of the table headings and legends according to the journal guidelines, ensuring consistency and improved readability. To present data more effectively, we have replaced Tables 2, 3, and 4 with corresponding figures (Figures 6, 7, and 8, respectively). This visual representation helps in better understanding the data and makes key differences more apparent. The original tables, which contain valuable detailed information, have been moved to the supplementary material (Tables S1, S2, and S3). Tables 1 and 5 from the original submission have been retained in the revised manuscript (as Table 1 and Table 2), with improved appearance and references to enhance clarity. The mean bacterial community compositions according to phylum in the Vag-pre, Vag-post, and Utr-post groups are shown in Figure 3 and detailed in the corresponding Table 1. Regarding Table 5 from the original manuscript (results of the DMM analysis), we considered converting it into a figure but found it challenging due to the large volume of data and the detailed nature of the information. We believe that transforming this detailed table into a figure might not effectively convey the complexity and nuances of the results. However, the conditional formatting included in all tables enhances the visual representation of the data, making key differences more apparent. These improvements, based on your feedback, have significantly enhanced the clarity and organization of our manuscript, facilitating better understanding and interpretation of the results.

_ _ _

Comment 3: There is no specific conclusion after the discussion, which is recommended to be added.

Response 3: We thank the reviewer for this comment and acknowledge it. While a summary paragraph was included at the end of the Discussion, and a separate conclusion section is not mandatory according to the journal guidelines, we agree that adding a dedicated Conclusions section, particularly when the discussion is unusually long or complex, is beneficial. Recognizing this, we have now added a dedicated Conclusions section to clearly highlight the key findings and their significance. This revised section succinctly summarizes the main outcomes and conclusions of our study, emphasizing the dynamic microbial transitions in the reproductive tract of first-pregnancy heifers from late gestation to early postpartum. We believe this addition enhances the manuscript by providing a clear and concise summary of the study's conclusions, as well as contributions and implications for future research.

_ _ _

Comment 4: The indexes of animal selection are too vague, please supplement the data, and it is suggested to supplement the indexes of serum immunity and inflammatory factor.

Response 4: We appreciate your suggestion to provide more detailed information on the indexes of animal selection. The inclusion and exclusion criteria for the animals in our study are thoroughly described in Section 5.1 of the Materials and Methods, and they were highlighted in the first paragraph of the Results section.

We did not analyze serum immunity and inflammatory factors in this study, as our primary focus was on the microbial transitions in the reproductive tract. The inclusion criteria ensured that the selected heifers were considered healthy and had no recent antibiotic treatments, which could influence microbial composition. The exclusion criteria aimed to eliminate potential confounding factors related to pregnancy complications and postpartum infections.

_ _ _

Comment 5: The materials and methods section introduces the classification of operational units according to the genus level, and the results section shows the classification at the phylum level, which is recommended to be checked.

Response 5: We appreciate the opportunity to clarify the taxonomic classification levels used in our study. We realized there were some mistakes in the headings of the tables, which caused some confusion, but have now been corrected. The Materials and Methods section details the bioinformatic analysis procedures, including the generation of operational taxonomic units (OTUs) and their classification. Specifically, the generation of OTUs was performed from phylotypes, which were classified at the genus level. This process ensures a high-resolution classification of microbial communities, allowing us to identify specific genera present in the samples. However, for the presentation of certain results, such as the overall composition of microbial communities and the comparison among groups (Vag-pre, Vag-post, and Utr-post), we also included taxonomic summaries at both the phylum and genus levels. This approach provides a broader perspective on the major microbial groups and their relative abundances, making it easier to identify significant shifts in microbial composition across different conditions. To clarify, the genus level was used for detailed taxonomic classification and analysis of specific OTUs, allowing for the identification of key microbial taxa distinguishing different groups, as described in the LEfSe analysis. The phylum and genus levels were used to present overall microbial composition and highlight major shifts in community structure across different sample groups. We hope this explanation clarifies the use of different taxonomic levels in our analysis and presentation of results.

_ _ _

Comment 6: According to the preface, the article intends that microorganisms are linked to inflammation or disease, whereas this article only does pre-work microbial analysis and does not clearly express whether or not it screens for key microorganisms associated with inflammation;

Response 6: Our study (new) title is "Reproductive Tract Microbial Transitions from Late Gestation to Early Postpartum Using 16S rRNA Metagenetic Profiling in First-Pregnancy Heifers”. Accordingly, our research aimed to describe and compare the microbial communities in the vagina at late gestation and in the vagina and uterus at early postpartum in first-pregnancy heifers, using 16S rRNA amplicon next-generation sequencing data. The heifers included in the study were selected based on specific criteria to ensure the focus and reliability of the results: they were first-pregnancy Holstein-Friesian heifers from a commercial dairy farm, without a history of pregnancy-related illness during gestation, and not treated with antibiotics six weeks prior to sampling. They were excluded if they had difficult assisted calving, twin pregnancy, retained fetal membranes, or postpartum septic metritis. While our study does not directly investigate uterine inflammatory diseases, it provides valuable foundational data on microbial transitions during this critical period. It addresses a significant gap in understanding the dynamics of microbial colonization, particularly in primiparous cows experiencing their first and most massive microbial invasion into a relatively naïve uterus. By employing next-generation sequencing, we comprehensively characterized the bacterial communities and their changes across anatomical sites and time points. The distinct profiles observed in the vaginal and uterine environments are crucial for future studies aiming to understand the increased susceptibility of primiparous females to postpartum uterine inflammatory diseases.

Our findings set the stage for future research to explore specific microorganisms' roles in health and disease, providing a valuable framework for developing targeted interventions to modulate microbial communities and improve reproductive outcomes in both veterinary and human medicine.

_ _ _

Comment 7: The table in the article is not conducive to observing the data and it is recommended to change it to a picture.

Response 7: As the original manuscript included five tables, it was unclear to us which specific table the reviewer refers to. Still, as detailed above, we have standardized the format of the table headings and legends according to the journal guidelines and replaced Tables 2, 3, and 4 with corresponding figures (Figures 6, 7, and 8) to enhance visual representation and clarity. Tables 1 and 5 have been retained in the revised manuscript as Table 1 and Table 2, with improved appearance and clearer references in the text.

Thank you!

Reviewer 3 Report

Comments and Suggestions for Authors

In the present work, Druker try to describe and compare the community compositions in the vagina at late gestation and in the vagina and uterus at early postpartum, in first-pregnancy heifers. This manuscript revealed distinct shifts in microbial composition between the prepartum vagina, postpartum vagina, and postpartum uterus. However, there are some questions that should be explained.

Major concerns

1. Why were samples collected from vaginal samples at gestation day 258±4, and vaginal and uterine samples at postpartum day 7±2.

2. The samples were collected from 33 first-pregnancy Holstein-Friesian heifers. We would like to know if 33 first-pregnancy Holstein-Friesian heifers delivery naturally or not, which had essential effects on the results.

3. The Holstein-Friesian heifers were at gestation day 258±4 and at postpartum day 7±2, and vaginas were under different endocrine environment, including progesterone. Therefore, the community compositions must be different. This reason should be considered.

Minor concerns

1. Abstract section should be rewritten. Lines 14-19 should be refined, and a summary sentence may be added in the end of the Abstract section.

2. Results section, in general, there is no reference Results section, and references 20-23 should be moved to Materials and Methods section.

3. The format of Tables should be revised.

4. References 1, 6, 8 are not the suitable to cite here in Line 288.

5. Lines 289-306, 314-332, 371-399, there is only one reference in these paragraphs.

6. Lines 413-429, as a conclusion paragraph, this should be refined.

7. Lines 432-433, delete ‘All animal handling and sampling procedures were performed by a certified veterinarian according to the ethical approval obtained from the Hebrew University  Institute Animal Care and Use Committee (IACUC MD-13-13807-2).’, which is repeated to Lines 609-611 .

8. Lines 470, 496, 514, ‘ºC,’ are different.

9. Line 594, ‘P’ should be in italic.

10. The format of some references is not suitable for this Journal. For example references 5, 29…

Comments on the Quality of English Language

Extensive editing of English language required.

Author Response

Dear Reviewer 3,             

Thank you very much for taking the time to review our manuscript and for providing positive feedback and valuable criticism. Based on your and the other reviewers' judgment and comments, the manuscript underwent substantial changes, including the Abstract, Introduction, descriptions of the Methods and Results, as well as relevant matching changes in the Discussion. Please find the detailed responses below and the revisions made. As part of our submission, two main manuscript files were uploaded - a clean copy (uploaded as the main manuscript file), and a copy with track changes (uploaded via Non-published Material).

We hope you will find the revised version suitable for publication in IJMS.

_ _ _

Comment 1: Why were samples collected from vaginal samples at gestation day 258±4, and vaginal and uterine samples at postpartum day 7±2.

Response 1: The samples were collected from vaginal samples at gestation day 258±4, and vaginal and uterine samples at postpartum day 7±2 to align with our study goals and considerations related to the gestation length of Holstein Friesian cows. The primary aim was to describe and compare the microbial community compositions in the vagina at late gestation and in the vagina and uterus at early postpartum in first-pregnancy heifers. The average gestation length for Holstein Friesian cows is approximately 280 days, with a normal range of about 270 to 290 days for most cows (or 265 to 295 days for all cows). Collecting samples at gestation day 258±4 ensures that we capture the microbial community in the vagina during late gestation, before the changes that might be associated with the preparation and onset of labor, establishing a baseline profile before the cervix opens and the uterus becomes susceptible to contamination. A sampling at postpartum day 7±2 allows us to observe the early stages of microbial colonization and community shifts following parturition, a period characterized by significant physiological changes, including uterine involution and new microbial establishment. This postpartum timing was also selected because several studies examine the uterine microbiome at this stage, allowing us and others to compare the results to previous studies.

We are aware that it would have been optimal to sample the cows more frequently (e.g., every other day, for a longer time) to provide more data, but this was beyond the scope and capability of this study. Still, the selected time points were chosen to ensure consistency and comparability across subjects, minimizing variability and capturing critical windows into microbial transitions. Our approach provides essential insights into microbial dynamics during these periods, which is crucial for understanding potential microbial markers or shifts associated with postpartum uterine health or disease, and highlights the need for future studies to expand on this work. Thanks to the reviewer's comments, we have added clarification and relevant references in the Discussion regarding the rationale for selecting these sampling days (see first paragraph in Discussion).

_ _ _

Comment 2: The samples were collected from 33 first-pregnancy Holstein-Friesian heifers. We would like to know if 33 first-pregnancy Holstein-Friesian heifers delivery naturally or not, which had essential effects on the results.

Response 2: Thank you for your valuable feedback. We have added clarification in the manuscript that all heifers included in the study calved by spontaneous vaginal delivery. This essential information has been incorporated into the relevant section of the text (Material and Methods; Results) to provide a clearer understanding of the study's inclusion criteria.

_ _ _

Comment 3: The Holstein-Friesian heifers were at gestation day 258±4 and at postpartum day 7±2, and vaginas were under different endocrine environment, including progesterone. Therefore, the community compositions must be different. This reason should be considered.

Response 3:  As suggested, we have revised the Discussion to consider possible endocrine differences and their effects on our results (see second paragraph and beyond for the revisions).

_ _ _

Comment 4: Abstract section should be rewritten. Lines 14-19 should be refined, and a summary sentence may be added in the end of the Abstract section.

Response 4: As recommended, the abstract was substantially revised.

_ _ _

Comment 5: Results section, in general, there is no reference Results section, and references 20-23 should be moved to Materials and Methods section.

Response 5: Thank you for your feedback regarding the Results section. According to the journal guidelines, the Results section precedes the Materials and Methods section. To facilitate reader understanding of the study, we have included brief and essential descriptions of the study design within the Results section where necessary. This includes a few references that are also cited in the Materials and Methods section to ensure clarity and coherence. Beyond these essential references, the Results section focuses solely on the findings of the current study, as additional references are not needed for presenting our results. In the Discussion section, we have included other references that are relevant for understanding our findings or for comparison to our results. We believe this approach effectively communicates the study's outcomes while maintaining adherence to the journal's structure and guidelines.

_ _ _

Comment 6: The format of Tables should be revised.

Response 7: Thank you for your feedback regarding the format of the tables. We have revised the format of the table headings and legends to standardize them according to the journal guidelines. Additionally, we have improved the appearance of the tables to enhance clarity and readability. Furthermore, to present data more effectively, we have replaced Tables 2, 3, and 4 with corresponding figures (Figures 6, 7, and 8) to provide a clearer visual representation of the results.

_ _ _

Comment 4: References 1, 6, 8 are not the suitable to cite here in Line 288.

Response 4:  Thank you for your feedback regarding the cited references. We believe that the references (Sheldon et al., 2008; Sicsic et al., 2018; Zhu et al., 2022) are relevant to the discussion on the microbial landscape and its implications for the postpartum uterus for the following reasons: Sheldon et al. (2008) discusses the bacterial contamination of the uterine lumen in cattle after parturition and its impact on uterine health, highlighting the role of some bacteria and the immune response in the postpartum uterus. This aligns with our findings on the lower richness and diversity in the postpartum uterus compared to the vagina, as well as the influence of luminal secretions and immune responses. Sicsic et al. (2018) compared the bacterial communities and inflammatory responses in the endometrium of healthy and metritic dairy cows postpartum, providing evidence that postpartum uterine microbial composition shows less diversity and richness in some cases. This aligns with our observations of the postpartum uterine environment being selective and less diverse. Zhu et al. (2022) highlighted the importance of microbial diversity and composition in the female reproductive tract and its impact on health, including during pregnancy. The findings support our discussion on the dynamic microbial landscape associated with parturition and the lower diversity observed in the postpartum uterus. We believe these references are suitable as they provide foundational and comparative context for our findings on the postpartum uterine microbial landscape and its implications. Additionally, due to the reviewer’s comment, we have added two relevant references: Fox et al. (2015) and DiGiulio et al. (2015). Fox et al. (2015) discusses the human microbiome associated with pregnancy outcomes, highlighting the significance of the microbiome in the female reproductive tract. DiGiulio et al. (2015) provides insights into the temporal and spatial variation of the microbiome in the reproductive tract during pregnancy and postpartum, supporting our discussion on microbial changes associated with parturition. Additionally, we have moved the references to a more appropriate location within the last sentence of this paragraph to avoid any confusion. We hope this clarifies the reviewer's concern.

_ _ _

Comment 5: Lines 289-306, 314-332, 371-399, there is only one reference in these paragraphs.

Response 5: Thank you for this critical comment. We have added several relevant references to these paragraphs to enhance the depth and support of our discussion.

_ _ _

Comment 6: Lines 413-429, as a conclusion paragraph, this should be refined.

Response 6: We thank the reviewer for this comment and acknowledge it. We have now added a dedicated Conclusions section to clearly highlight the key findings and their significance. This revised section succinctly summarizes the main outcomes and conclusions of our study, emphasizing the dynamic microbial transitions in the reproductive tract of first-pregnancy heifers from late gestation to early postpartum. We believe this addition enhances the manuscript by providing a clear and concise summary of the study's conclusions, as well as contributions and implications for future research.

_ _ _

Comment 7: Lines 432-433, delete ‘All animal handling and sampling procedures were performed by a certified veterinarian according to the ethical approval obtained from the Hebrew University  Institute Animal Care and Use Committee (IACUC MD-13-13807-2).’, which is repeated to Lines 609-611 .

Response 7: We have revised this statement to fulfill the IJMS journal requirements in this section.

_ _ _

Comment 8:. Lines 470, 496, 514, ‘ºC,’ are different

Response 8: The format presentation was corrected as indicated.

_ _ _

Comment 9: Line 594, ‘P’ should be in italic.

Response 9: The format presentation of  P value was corrected throughout the manuscript as instructed by the reviewer.

_ _ _

Comment 10: The format of some references is not suitable for this Journal. For example references 5, 29….

Response 10: All references were inserted uniformly using EndNote to meet the journal's specific requirements.

_ _ _

Comment 11: Comments on the Quality of English Language: Extensive editing of English language required.

Response 11: The manuscript was edited by native English-speaking researchers and also by English editing software.

Thank you!

Reviewer 4 Report

Comments and Suggestions for Authors

The work "Reproductive Tract Microbial Transitions in First-Pregnancy Heifers: 16S rRNA Metagenetic Profiling of Vaginal Microbiota in Late Gestation and Vaginal and Uterine Microbiota in Early Postpartum" aimed to describe and compare the community compositions in the vagina at late gestation and in the vagina and uterus at early postpartum, in first-pregnancy heifers using r16S amplicon NGS data. The manuscript is generally well-written. The main concern with the work is that it would be much more suitable for a veterinary-oriented journal.

Minor suggestions: The abstract is not written according to scientific standards. It should be shortened and written according to the abstract sections.

The conclusion section is missing.

What is the main scientific contribution of the study? Where should obtained results be implemented? Kindly additionally explain where could obtained results improve certain outcomes.

Author Response

Dear Reviewer 4,              

Thank you very much for taking the time to review our manuscript and for providing positive and valuable feedback. Based on your and the other reviewers' judgment and comments, the manuscript underwent substantial changes, including the Abstract, Introduction, descriptions of the Methods and Results, as well as relevant matching changes in the Discussion. Please find the detailed responses below and the revisions made. As part of our submission, two main manuscript files were uploaded - a clean copy (uploaded as the main manuscript file), and a copy with track changes (uploaded via Non-published Material).

We hope you will find the revised version suitable for publication in IJMS.

_ _ _

Comment 1: The work "Reproductive Tract Microbial Transitions in First-Pregnancy Heifers: 16S rRNA Metagenetic Profiling of Vaginal Microbiota in Late Gestation and Vaginal and Uterine Microbiota in Early Postpartum" aimed to describe and compare the community compositions in the vagina at late gestation and in the vagina and uterus at early postpartum, in first-pregnancy heifers using r16S amplicon NGS data. The manuscript is generally well-written. The main concern with the work is that it would be much more suitable for a veterinary-oriented journal.

Response 1: We appreciate your positive feedback on the quality of the manuscript. While our study focuses on first-pregnancy heifers, the insights gained extend beyond veterinary applications and contribute to the broader field of reproductive biology and microbial ecology. Understanding microbial transitions in the reproductive tract is crucial for both veterinary and human medicine, as these findings can inform practices aimed at improving reproductive health and managing infections. Thus, we believe our study is well-suited for the interdisciplinary audience of IJMS. We hope you will agree with us after reviewing the revised manuscript.

_ _ _

Comment 2: The abstract is not written according to scientific standards. It should be shortened and written according to the abstract sections.

Response 2: Thank you for your feedback. We have revised the abstract to ensure it adheres to scientific standards, making it structured according to typical abstract sections, including background, objectives, methods, results, and conclusions.

_ _ _

Comment 3: The conclusion section is missing.

Response 3: Thank you for pointing this out. We have added a conclusion section to the manuscript. The new conclusion section summarizes the key findings, highlights the significance of our research, and suggests directions for future studies.

_ _ _

Comment 4: What is the main scientific contribution of the study? Where should obtained results be implemented? Kindly additionally explain where could obtained results improve certain outcomes.

Response 4: The main scientific contribution of our study is the detailed characterization of microbial transitions in the reproductive tract of first-pregnancy heifers from late gestation to early postpartum. By utilizing 16S rRNA metagenetic profiling, we identified significant shifts in microbial composition and diversity. These findings enhance our understanding of the reproductive tract microbiome's role in health and disease, particularly in primiparous females who are at a higher risk of postpartum uterine inflammatory diseases.

The obtained results can be implemented in several ways. In the field of reproductive biology research, the study provides a foundation for further investigation into the mechanisms underlying microbial transitions in the reproductive tract, potentially leading to novel therapeutic approaches for managing reproductive health in both animals and humans. In veterinary practice, our findings can help veterinarians develop better management strategies to prevent and treat postpartum infections in dairy cows, thereby improving animal welfare and productivity. Additionally, understanding the dynamics of microbial communities in the reproductive tract can inform broader ecological studies on microbial interactions and their impact on host health.

Overall, our results can lead to improvements in reproductive health outcomes by informing targeted interventions and management practices in both veterinary and human medicine. The revised manuscript highlights the possible explanations for our findings, relevant impacts, and future directions. We therefore hope you will find it interesting, relevant, and valuable to the scientific community.

Thank you!

Round 2

Reviewer 2 Report

Comments and Suggestions for Authors

1The conclusion part is a summary of the research results, as well as future prospects, etc. It is recommended that the authors streamline and revise the conclusion part again ;

2Please check the number scale in the Materials and Methods section;

3The temperature format of 581 lines is wrong, please change it; it is suggested to check the whole text again.

Author Response

Dear Reviewer 2,

Thank you very much for taking the time to review our revised manuscript and for providing positive feedback and valuable criticism. We have revised the manuscript based on your and the other reviewers' judgment and comments.

We hope you will find the revised version suitable for publication in IJMS.

Comment 1: The conclusion part is a summary of the research results, as well as future prospects, etc. It is recommended that the authors streamline and revise the conclusion part again ;

Response 1: Thank you for your valuable feedback. We have streamlined and revised the conclusions section. We hope this addresses your concerns and enhances the clarity of our manuscript.

- - -

Comment 2: Please check the number scale in the Materials and Methods section;

Response 2: Upon careful review, we have ensured that the number scale and all numerical data are presented clearly and consistently. If there are any particular sections or numerical data points you believe need further clarification, please let us know, and we will address them promptly.

- - -

Comment 3: The temperature format of 581 lines is wrong, please change it; it is suggested to check the whole text again.

Response 3: Thank you for your feedback. We have corrected the temperature format throughout the manuscript and ensured consistency.

- - - 

Thank you!

Reviewer 3 Report

Comments and Suggestions for Authors

Thanks for author’s responses. However, extensive editing of English language required. For example,

1. Line 7, ‘3 , *’, delete ‘,’

2. Line 22, ‘heifers and women remain limited*’, delete ‘and women’. This study is not involved in women.

Comments on the Quality of English Language

Extensive editing of English language required.

Author Response

Dear Reviewer 3,

Thank you very much for taking the time to review our revised manuscript and for providing positive feedback and valuable criticism. We have revised the manuscript based on your and the other reviewers' judgment and comments.

We hope you will find the revised version suitable for publication in IJMS.

Comment 1: Thanks for author's responses. However, extensive editing of English language required. For example,  1. Line 7, '3 , *', delete ','

Response 1: Thank you for your feedback. The manuscript has been thoroughly edited for proper English. If the reviewer has any specific concerns, please inform us, and we will be happy to address them. Regarding the particular example (#1), the comma between the number (3) and the asterisk (*) is needed based on the journal guidelines, as the latter indicates the corresponding author. The excess spaces between numbers indicating affiliations have been deleted as suggested.

- - -

Comment 2:  Line 22, 'heifers and women remain limited*', delete 'and women'. This study is not involved in women.

Response 2: Thank you for this comment. We understand the importance of accurately reflecting the scope of our study, and we acknowledge that our original sentence, mentioning women specifically, was potentially misleading. While our research specifically focuses on first-pregnancy heifers, the broader context of limited research on spatio-temporal microbial transitions in the reproductive tract applies to many species. Therefore, we revised the sentence to be more generalized, emphasizing this knowledge gap across species. The revised sentence now reads: "However, investigations into spatio-temporal microbial transitions in the reproductive tract of primigravid females remain limited." This change highlights the broader relevance of our findings and the need for further research in this area. Additionally, the following sentence clearly indicates that the study focuses on first-pregnancy heifers, ensuring clarity and precision in our manuscript.

- - -

Thank you!

Reviewer 4 Report

Comments and Suggestions for Authors

Authors have made improvements to the manuscript but the main issue remains that it would be more suitable for an animal-oriented/veterinary journal.

Comments on the Quality of English Language

Minor editing necessary

Author Response

Dear Reviewer 4,

Thank you very much for taking the time to review our revised manuscript and for providing positive feedback and valuable criticism. We have revised the manuscript based on your and the other reviewers' judgment and comments.

We hope you will find the revised version suitable for publication in IJMS.

Comment 1: Authors have made improvements to the manuscript but the main issue remains that it would be more suitable for an animal-oriented/veterinary journal.

Response 1: Thank you for your feedback and recognition of the improvements made to our manuscript. While our study indeed focuses on first-pregnancy heifers, the findings have broader implications that extend beyond veterinary science. The microbial transitions observed in the reproductive tract have significant relevance to reproductive biology and microbial ecology, fields that are of interest to the broader scientific community, including those studying similar processes in other animals and humans. We believe that the insights gained from this research can inform and enhance understanding of reproductive health and microbial dynamics across different species, making it suitable for publication in IJMS, which encompasses a wide range of biological and medical sciences. We hope you will consider this broader perspective in evaluating the suitability of our revised manuscript for this journal.

- - -

Comment 2: Comments on the Quality of English Language. Minor editing necessary

Response 2: The manuscript has been thoroughly edited for proper English. If the reviewer has any specific concerns, please inform us, and we will be happy to address them.

- - -

Thank you!

Round 3

Reviewer 4 Report

Comments and Suggestions for Authors

Dear Authors,

thank you or your kind reply. The scientific paper would indeed be more suitable for a veterinary-oriented journal.

Wishing you all the best.

Comments on the Quality of English Language

minor editing needed